

# Ground-based Temperature and Humidity Profile Retrieval Using Infrared Hyperspectrum Based on Adaptive Fast Iterative Algorithm

Wei Huang[1], Lei Liu[2], Bin Yang[1], Shuai Hu[2], Wanying Yang[2], Zhenfeng Li[1], Wantong Li[3], Xiaofan Yang[1]

[1]The State Key Laboratory of Complex Electromagnetic Environment Effects on Electronic and Information System, Luoyang 471003, China

[2]College of Meteorology and Oceanography, National University of Defense Technology, Changsha 410073, China

[3]Tianjin Meteorological Radar Research & Trial Centre, Tianjin 300061, China

*Correspondence to:* Lei Liu (liulei17c@nudt.edu.cn)

**Abstract.** Due to the complex radiative transfer process, the retrieval time of the physical retrieval algorithm is significantly increased compared with that of the statistical retrieval algorithm. The calculation of the Jacobian matrix is the most computationally intensive part of the physical retrieval algorithm. Further analysis showed that the changes in Jacobians had little effect on the performance of the physical retrieval algorithm. On the basis of the above findings, a fast physical-iterative retrieval algorithm was proposed by adaptively updating the Jacobian in keeping with the changes of the atmospheric state. The performance of the algorithm is evaluated using synthetic ground-based infrared spectra observations. The retrieval speed is significantly improved compared with the traditional physical retrieval algorithm under the condition that the parameters of the computing platform remain unchanged, with the average retrieval time reduced from 8.96 min to 3.69 min. The retrieval accuracy of the fast retrieval model is equivalent to that of the traditional algorithm, with maximum root-mean-square errors of less than 1.2 K and 1.0 g/kg for heights below 3 km for the temperature and water vapor mixing ratio (WVMR), respectively. The Jacobian updating strategy has a certain impact on the convergence of the retrieval algorithm, whose convergence rate is 98.7%, which is lower than that of the traditional algorithm to some extent. However, reliable retrieval results can still be obtained by adjusting the convergence criteria.

## 1 Introduction

High-quality profiles of atmospheric constituents are required for many endeavors, including radiative transfer, cloud process research, and assimilation into mesoscale models to improve forecasts (Turner et al., 2000). The accuracy of the initial field provided by the observation network is becoming a key factor restricting the skill of the NWP model (Romine et al., 2013; Li et al., 2016). The existing observation network is insufficient to meet the needs of the convective scale numerical weather prediction system, especially in the prediction of convection initiation convective processes (Kain et al., 2013; Wagner et al., 2019; Geerts et al., 2018). As the spatiotemporal resolution is too coarse, the radiosonde profiles cannot capture the



atmospheric phenomena in detail. Space-based detection equipment observes atmospheric upwelling radiance, which shows

some drawbacks in the detection of the planetary boundary layer (PBL) owing to the influence of the cloud layer and

underlying surface. A promising solution is the ground-based infrared spectra detection platform, which shows more

advantages in retrieving the temperature and humidity profiles of the PBL compared with the space-based detection platform

by observing infrared hyperspectrum in a downward way. The assimilation of ground-based infrared hyperspectral data can

significantly improve the ability of the convective scale prediction system for convection initiation (Coniglio et al., 2019; Hu

et al., 2019).

The commonly used ground-based infrared hyperspectral equipment mainly includes Fourier Transform Infrared (FTIR)

instruments of the Karlsruhe Institute of Technology deployed in the Detection of Atmospheric Composition Change

(NDACC) (De Mazière et al., 2018) and Atmospheric Emitted Radiance Interferometer (AERI) developed by the University

of Wisconsin Space Science and Engineering Center (UW-SSEC) deployed in the Atmospheric Radiation Measurement

(Knuteson et al., 2004). FTIR observes near-infrared and mid-infrared high-resolution solar spectra, which are mainly used

to retrieve water vapor (Schneider et al., 2006a, b; Schneider and Hase, 2009), water isotopologues (Schneider et al., 2006a;

Barthlott et al., 2017) and various trace gas (Gardiner et al., 2008; Kiel et al., 2016; Zhou et al., 2018; Yin et al., 2020; Yin et

al., 2021a; Yin et al., 2021b) profiles. The spectral region of AERI covers the range of 520-3000 $cm^{-1}$, containing a 15 µm

absorption band of $CO_2$ commonly used for the retrieval of temperature profiles, which is more advantageous in detecting

thermodynamic profiles. Specific retrieval algorithms, capable of being divided into statistical retrieval algorithms and

physical retrieval algorithms as per different principles, are required to extract large amounts of information on the required

atmospheric profiles from rich infrared hyperspectral radiance data. The physical retrieval algorithm considers the interaction

process between electromagnetic waves and atmospheric constituents, which enables it to provide thermodynamic profiles

with higher accuracy than the statistical retrieval algorithm (Yang and Min, 2018; Cimini et al., 2010). AERI equipment has

successively adopted two physical retrieval schemes, named AERIprof (Smith et al., 1999; Feltz et al., 1998) and AERIoe

(Turner and Löhnert, 2014; Turner and Blumberg, 2019; Turner and Löhnert, 2021). Based on the "onion peeling" algorithm,

the former is used to adjust the first-guess profile from bottom to top with the iterative algorithm as per the difference

between the simulated radiation and the observed radiation. Given that the algorithm only needs to calculate the diagonal

elements in the Jacobian matrix, its retrieval speed is faster than that of the optimal estimation method (OEM) (Rodgers,

2000).

However, the AERIprof algorithm has some outstanding shortcomings to boot, mainly including that it not only is

greatly affected by the first-guess profile but also cannot provide uncertainty on retrieval results (Turner and Löhnert, 2014;

Blumberg et al., 2017; Blumberg et al., 2015). The limitations of AERIprof could be overcome by the AERIoe

optimal-estimation retrieval algorithm, which was designed as an alternative to the previous physical algorithm. One of the important improvements remains to reduce the dependence on the first-guess profile by introducing regularization parameters in the AERIoe algorithm to adjust the balance between the observation information and the background field. The AERIoe algorithm sets the regularization parameters as fixed values from large to small to achieve good stability and accuracy, which makes the algorithm require at least 7 iterations. The Jacobian matrix should be updated for each iteration

due to the dependence on the current state vector, which significantly increases the amount of calculation and results in a high retrieval time.

A fast physical-iterative retrieval method, henceforth called Fast AERIoe, is proposed aiming at the problem of long retrieval time in the AERIoe. The computation amount is reduced by adjusting the updating strategy of the Jacobian, which can improve the retrieval speed of AERIoe; the Jacobians, by monitoring the index of the iterative profiles, can be updated

adaptively without manual intervention. Finally, the retrieval time, convergence characteristics and accuracy of the new algorithm are presented using radiosonde data at the same station.

## 2 Data

The data used in the research are from ARM program supported by the U. S. Department of Energy, which aims to quantitatively study the atmospheric radiation budget and develop and verify the parameterization scheme of the numerical

model (Revercomb et al., 2003; Ellingson et al., 2016). This program mainly focuses on the long-term observation of atmospheric states and radiative fluxes, providing information to researchers around the world to inform and validate predictive models of climate and weather. We will use data collected at the Southern Great Plain (SGP) site, which is located at 36.61 ° N and 149.88 ° W, near Lamont, Oklahoma, USA (Sisterson et al., 2016). These data mainly include ground-based infrared spectra obtained by AERI and radiosonde profiles, with the former used to retrieve the temperature and water vapor

profiles and the latter mainly used to evaluate the accuracy of the retrieval results.

### 2.1 AERI

AERI can continuously receive downwelling atmospheric infrared radiance from 3.3-19.2 μm (520-3000 cm$^{-1}$) with a spectral resolution better than 1 cm$^{-1}$, among which the infrared radiation of the 520-1800 cm$^{-1}$ band is obtained by the mercury cadmium telluride (HgCdTe) detector, with 1800-3020 cm$^{-1}$ band obtained by the indiumantimonide (InSb) detector.

The AERI front-end optics include a scene mirror and two calibrated blackbodies, one of which changes with the temperature of the surrounding environment, while the other maintains a fixed temperature (60 ℃). AERI achieved a calibration accuracy of better than 1% by viewing two high-precision blackbodies and a nonlinearity correction for the



detectors (Knuteson et al., 2004). The temporal resolution of the AERI standard remains approximately 8 minutes, including a 3-minute sky dwell period and 5-minute periods for each of the blackbodies.

**Table 1.** Spectral regions used for retrieving temperature and WVMR profiles in the AERIoe algorithm

| Temperature | Water Vapor |
|---|---|
| 612-618 cm$^{-1}$ | 538-588 cm$^{-1}$ |
| 624-660 cm$^{-1}$ | |
| 674-713 cm$^{-1}$ | |

AERI has many observation channels, including not only temperature and humidity profile information but also trace gas information such as ozone, methane, and redundant data. Therefore, appropriate channels must be selected when retrieving temperature and humidity profiles. The retrieval of humidity profiles generally adopts water vapor-sensitive channels, and the retrieval of temperature profiles generally adopts the sensitive band of gas composition (such as $CO_2$) with stable content. The channels used in the retrieval process are consistent with AERIoe v1.2, which used only the 538–588 cm$^{-1}$ spectral region for water vapor profiling to exclude scattering effects from clouds (Turner and Blumberg, 2019). The specific frequencies are shown in Table 1, among which the spectral region used for temperature retrieval includes 167 channels, and the water vapor includes 104 channels.

**2.2 Radiosonde data**

Radiosondes have been used for decades to provide humidity, temperature and wind profiles throughout the troposphere, which is considered to be the most accurate means to detect the vertical structure of the atmosphere. It is often used to evaluate the accuracy of other detection methods. Located 150 m to the north of the AERI equipment, the closer radiosonde release point can ensure the comparability of the radiosonde profiles and AERI retrieval results (Wakefield et al., 2021). The radiosonde data at the SGP site were obtained by Vaisala RS92 since 2002 (Turner et al., 2016), including temperature, humidity, pressure, wind direction and wind speed. It was regularly launched four times a day at 05:30 UTC, 11:30 UTC, 17:30 UTC and 23:30 UTC.

We collected the radiosonde profiles and AERI radiation data of 2012, screening 826 groups of qualified data samples through quality control, spatiotemporal matching, and clear sky recognition. On the basis of the above datasets, we calculated the simulated AERI spectrum corresponding to 826 sets of radiosonde profiles using the LBLRTM, with parameter settings consistent with Sect. 3.1.



## 3 Methodology

The AERIoe algorithm, based on the optimal estimation method, comprehensively considers the observation information and atmospheric prior information, iteratively searching for the atmospheric state that most conforms to the observation and prior constraints.


$$\mathbf{X}_{n+1} = \mathbf{X}_0 + \left( \mathbf{K}_n^T \mathbf{S}_e^{-1} \mathbf{K}_n + \gamma \mathbf{S}_a^{-1} \right)^{-1} \mathbf{K}_n^T \mathbf{S}_e^{-1} \times \left( \mathbf{Y}^m - F\left( \mathbf{X}_n \right) + \mathbf{K}_n \left( \mathbf{X}_n - \mathbf{X}_0 \right) \right), \tag{1}$$

Here, $\mathbf{X}$ is the profile of the atmospheric state to be retrieved, $\mathbf{X}_0$ is the first-guess profile of the atmosphere, $\mathbf{Y}^m$ is the observed radiance vector, $F(\mathbf{X})$ is the AERI observed spectrum, $\mathbf{S}_e$ is the observation error covariance matrix, $\mathbf{S}_a$ is the background covariance matrix, and $n$ represents the number of iterations. The superscripts $T$ and -1 imply the matrix transpose and inverse, respectively.

To improve the stability of the retrieval algorithm, the regularization parameter $\gamma$ was introduced in Formula (1), which is set as 10 fixed values from large to small ([1000, 300, 100, 30, 10, 3, 10, 1]). As $\gamma$ decreases with iterations progress, more observation information is introduced to improve the retrieval accuracy. The retrieval is not allowed to converge until $\gamma$ decreases to 1 and the following convergence criterion is satisfied.

$$convergence\_index = \frac{(\mathbf{X}^n - \mathbf{X}^{n+1})\mathbf{S}^{-1}(\mathbf{X}^n - \mathbf{X}^{n+1})}{N} \le 1, \tag{2}$$

$N$ represents the dimension of the retrieved atmospheric state vector.

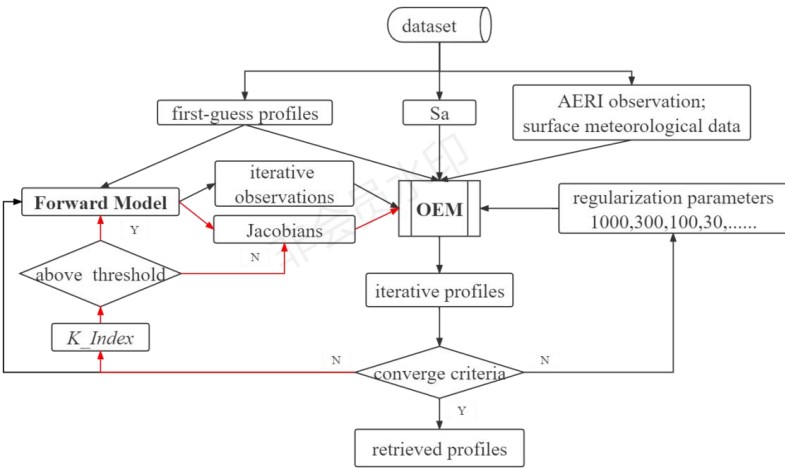

**Figure 1.** Flowchart of the Fast AERIoe retrieval process. Note that the red line indicates the Jacobian updating process.

The updating of the Jacobians in the above retrieval process requires the calculation of the optical thickness or radiation intensity of different atmospheric constituents at each height. On the condition that the Jacobian is calculated for each



iteration in the retrieval process, the calculation amount and time cost of retrieval will remain high. Owing to the constraints

of $\gamma$, the decrease of the difference between simulated and observed radiation is not very much in the adjustment of

individual iterations to the retrieval profile. At this time, the change in the Jacobian calculated as per the iteration profile is

negligible. Backed by the above analysis, a fast iterative algorithm called Fast AERIoe is proposed on the basis of the

AERIoe algorithm. The Jacobian is not updated to improve the retrieval speed of the algorithm when the adjustment of the

iterative profile is small. The retrieval process is shown in Fig. 1, which mainly includes the establishment of the background

field, the calculation of the forward model, and the adaptive iteration of the Jacobians.

### 3.1 Establishment of background field

The atmospheric state vector in retrieval includes temperature and water vapor. The retrieval form of water vapor profile is

different from that of temperature profile, as such it needs to be set in logarithmic form in the state vector. Relevant studies

show that the probability density function (*pdf*) of water vapor in the atmosphere does not meet the normal distribution, with

direct retrieval of water vapor resulting in smaller retrieval results (Schneider et al., 2006b). A normal *pdf* can be completely

described by its covariance and its mean when the water vapor in the state vector is transformed to a logarithmic scale, and

the solution is the maximum value of the conditional *pdf*. In addition, taking the logarithm of water vapor can avoid the

negative value of the retrieved water vapor profile (Cimini et al., 2011). As such, it is more reasonable to take the logarithm

of the water vapor profile in the state vector.

The retrieval height is related to the information content of the observation vector. As AERI radiation information is

mainly concentrated in the PBL, the highest retrieval height is set as 3 km, with the vertical resolution at the bottom set as

approximately 10 m (Turner and Löhnert, 2014). With increasing height, the amount of information content gradually

decreases. Thus, the vertical resolution of the retrieval profile should also increase accordingly, with the resolution at a

height of 3 km 300 m, and the entire height range divided into 37 layers. Since the retrieval algorithm only considers the

error correlation in the vertical direction, the background error covariance matrix $\mathbf{S}_a$ can be expressed as follows:

$$\mathbf{S}_a = \begin{pmatrix} \mathbf{S}_{a_{11}} & \cdots & \mathbf{S}_{a_{1j}} \\ \vdots & \ddots & \vdots \\ \mathbf{S}_{a_{i1}} & \cdots & \mathbf{S}_{a_{ij}} \end{pmatrix}, \tag{3}$$

$$\mathbf{S}_{a_{ij}} = cov\left(\mathbf{X}^i, \mathbf{X}^j\right) = \frac{1}{M}\sum_{k=1}^{M}\left[\left(\mathbf{X}_k^i - E\left(\mathbf{X}^i\right)\right)\cdot\left(\mathbf{X}_k^j - E\left(\mathbf{X}^j\right)\right)\right], \tag{4}$$

In the above formula, *M* is the number of samples. When the temperature and humidity fields from the NWP model are

used as the first-guess profile, **X** is the deviation of the atmospheric temperature and humidity profile between the NWP and





the radiosonde profile. When the statistical average values of the radiosonde profile are used as the first-guess profile, **X** can be simplified as the temperature and humidity profile. In this paper, the latter method is used in the establishment of $\mathbf{S}_a$, which does not need numerical prediction information and has less dependence on the NWP model. Fig. 2 shows $\mathbf{S}_a$ calculated from 826 sets of radiosonde profiles under clear skies in 2012. The poor initial profile witnesses significantly

larger $\mathbf{S}_a$ in the figure than that with the prediction field as the initial value (Wang et al., 2014; Guan et al., 2019), which significantly increases the instability of the retrieval algorithm. Regularization parameters must be used to impose constraints with a steeper descent of the cost function (Hewison, 2007). This is an important reason for the AERIoe algorithm to rely on the regularization parameter, with a key to reducing its dependence on the first-guess profile.

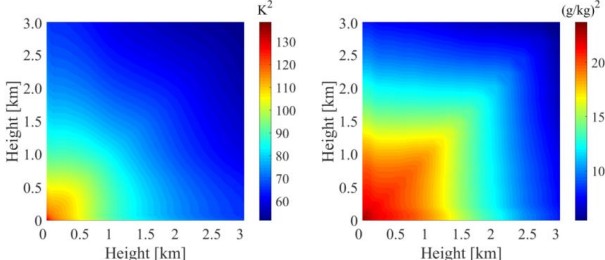

**Figure 2.** Temperature profile (left) and WVMR profile (right) background error covariance matrix.

**3.2 Forward model settings**

The construction of the forward model represents the most important part of the retrieval algorithm. It is an important bridge between the state vector represented by atmospheric temperature and humidity and the observed radiance and other observation vectors, determining the accuracy of the deviation between the observed radiance and the simulated radiance in

the iterative algorithm. In addition to the observed radiances of 271 channels in the inversion process, the observation vector **Y** also contains ground temperature and humidity observation data to improve the information content near the ground, as shown in Formula (5):

$$\mathbf{Y} = \left[\mathbf{R}_1, \mathbf{R}_2, \dots, \mathbf{R}_{271}, \mathbf{T}_{surf}, \ln \mathbf{q}_{surf}\right], \tag{5}$$

The forward model of surface meteorological observations remains relatively simple, only needing to convert its unit

into the atmospheric state vector, with the corresponding Jacobian matrix calculated by the finite differences method. The most accurate forward model for ground-based infrared hyperspectral radiation remains the Line-by-Line Radiative Transfer Model (LBLRTM) developed by the Atmospheric and Environmental Research (AER) Inc. using the line-by-line approach, which has been extensively validated against atmospheric radiance spectra from the ultraviolet to the submillimeter (Turner et al., 2004). The algorithmic accuracy of the LBLRTM is approximately 0.5%, and the errors associated with the



computational procedures are on the order of five times less than those associated with the line parameters (Clough et al., 2005). The Fast Fourier Transform (FFT) calculation module included in the LBLRTM accurately models the spectral response function by simulating the operation process of the Fourier Transform Spectrometer (FTS), which is more accurate and more efficient than the convolution technique. The comparative observation test with AERI equipment has promoted the continuous development of AERI hardware, highlighting the accuracy of LBLRTM calculation (Mlawer and Turner, 2016).

The LBLRTM model calculates the downwelling infrared radiation received by AERI as per the given atmospheric temperature and humidity profile, trace gas concentration profile and surface characteristic parameters, combined with the spectral response function of the hyperspectral sensor channel. During calculation, the atmospheric constituents to be set mainly include temperature, water vapor, $CO_2$, $O_3$, $CH_4$ and $N_2O$, of which temperature and water vapor represent the state vector to be retrieved and are set as the first-guess profile. $CO_2$, $CH_4$ and $N_2O$ are independent of altitude, with the content

set to 384, 1.793 and 0.310 ppm, respectively, with reference to the AERIoe algorithm (Turner and Löhnert, 2014). The concentration profile of $O_3$ is taken from the *U. S. Standard Atmosphere 1976*. LBLRTM also includes the Jacobian matrix calculation module, which can be used to derive the Jacobian of temperature, water vapor, and various trace gases in an analytical way. Compared with the finite difference method, the analytical Jacobian is more efficient and less affected by numerical error and nonlinear contributions (Clough et al., 2006).

**3.3 Adaptive updating of Jacobian**

Adaptive updating of Jacobian remains a key link of Fast AERIoe, determined whether updating or not by monitoring the indicators that can reflect the changes of Jacobian in the iterative process. Its establishment process mainly includes the following three aspects.

**3.3.1 Quantification of algorithm retrieval capability**

The retrieval accuracy of the atmospheric profile depends on the amount of atmospheric information in the hyperspectral data. Information Content (*IC*) and Degrees of Freedom for Signal (*DFS*), as important indicators to describe the detector's retrieval ability for specific atmospheric parameters, can quantitatively describe the effective information contained in hyperspectral data (Rodgers, 1998). *IC* represents the reduction in uncertainty in observation information caused by the retrieval process, with the calculation formula shown in (6). *DFS* represents the independent information contained in the

measured radiation, with the calculation formula shown in (7).

$$IC = \frac{1}{2} ln\, det\left(\hat{\mathbf{S}}^{-1}\mathbf{S}_a\right),$$ (6)

$$DFS = Trace\left(\mathbf{B}^{-1}\mathbf{K}_n^T\mathbf{S}_e^{-1}\mathbf{K}_n\right),$$ (7)



Here, $\hat{\mathbf{S}}$ is the posterior error covariance matrix, also known as the analysis error covariance matrix. Its diagonal element is the standard deviation of the retrieval error, with the calculation formula $\hat{\mathbf{S}}$ as follows:


$$\hat{\mathbf{S}} = \mathbf{B}^{-1}\left(\mathbf{K}^T\mathbf{S}_e^{-1}\mathbf{K} + \gamma^2\mathbf{S}_a^{-1}\right)\mathbf{B}^{-1}, \tag{8}$$

Among which,

$$\mathbf{B} = \left(\gamma\mathbf{S}_a^{-1} + \mathbf{K}_n^T\mathbf{S}_e^{-1}\mathbf{K}_n\right), \tag{9}$$

### 3.3.2 Monitoring index design

It can be seen from Formulas (6) and (7) that $\mathbf{S}_a$ and $\mathbf{S}_e$ are fixed during retrieval. What affects *IC* and *DFS* lies in only $\gamma$

and Jacobian. Owing to the difficulty of quantifying the change in the two-dimensional Jacobian caused by the iteration profiles, a monitoring index, henceforth called *K_Index*, is designed and used to characterize the change in the profiles at various iterations. The calculation of *K_Index* comes from the convergence criteria *convergence_index*, which contains not only the difference between the iteration profiles but also the posterior dominated by Jacobian. The introduced *K_Index* should reflect the changes in the temperature and humidity profile, which means that the influence of the Jacobian should not

be included. Then, the *convergence_index* was degenerated into the *K_Index* as follows.

$$K\_Index = \frac{\left(\mathbf{X}^n - \mathbf{X}^{n+1}\right)^T\left(\mathbf{X}^n - \mathbf{X}^{n+1}\right)}{N}, \tag{10}$$

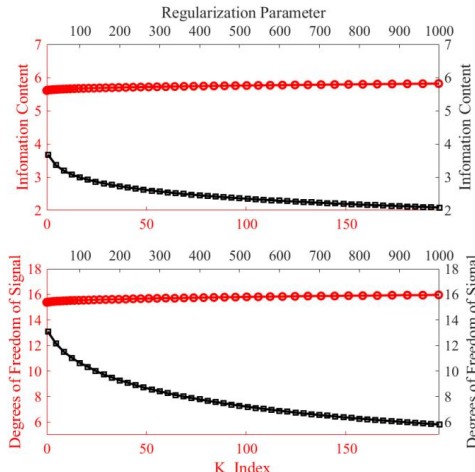

**Figure 3.** *IC* (top row) and *DFS* (bottom row) change with $\gamma$ (red lines with circles) and *K_Index* (black lines with squares).

The values of $\gamma$ and the *K_Index* will change with the adjustment of the profile during the retrieval process. Fig. 3

shows the curve of *IC* and *DFS* changed with $\gamma$ and *K_Index*. When $\gamma$ decreases from 1000 to 1, *DFS* and *IC* increases



by approximately 80% and 120%, respectively. The *K_Index* metric has the opposite effect compared to $\gamma$, with the value of SIC and *DFS* in the retrieval process proportional to the *K_Index*. However, the influence of *K_Index* is far less than that of $\gamma$, with the change range of *DFS* within 4% and *IC* within 5%. This shows that the change of Jacobian matrix has less influence on the retrieval ability than $\gamma$, which provides an effective means to improve the retrieval speed of AERIoe by

reducing the update frequency of the Jacobian. This could be achieved by comparing the relative size of the *K_Index* and its threshold of each iteration to determine whether the Jacobian is updated or not.

### 3.3.3 Determination of the K_Index threshold

The selection of the threshold for *K_Index* is very important for the Fast AERIoe algorithm: if the threshold remains too large, too many Jacobians will stop updating, resulting in the decline of retrieval results or even the convergence of the

retrieval process; while the threshold value remains too small, and most Jacobians need to be updated, which cannot effectively shorten the retrieval time.

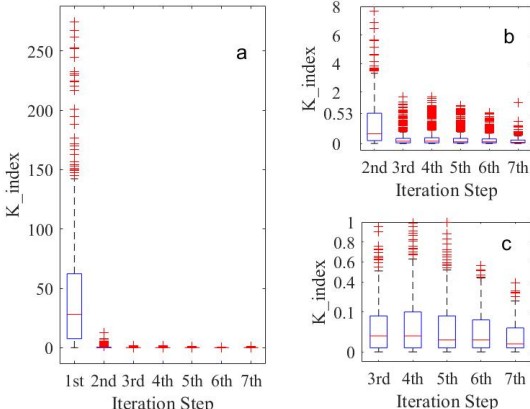

**Figure 4.** Box-and-whisker plots for *K_Index* values at different iterations in the retrieval process of AERIoe. (a) *K_Index* values calculated using 826 samples at iterations 1-7, (b) and (c) are same as (a), but for iterations 2-7 and iterations 3-7, respectively. The boxes

show upper-quartile, median (the red line through the middle of the box), and lower-quartile values for *K_Index*. The whiskers extend to the 1.5 times the inter quartile range (IQR). Any outliers above or below the whiskers are plotted as red symbols '+'.

Fig. 4 shows the histogram of the *K_Index* distribution for each iteration in the retrieval process, with the *K_Index* values at each iteration calculated using the clear sky data for 2012. Since the historical average profile was used as the first-guess, which has a large deviation from the real atmospheric state, a larger value of *K_Index* was demonstrated in the

first step of the retrieval. The *K_Index* value decreases significantly from the second iteration (see Fig. 4a), indicating that the adjustment of the iterative profile remains very small and the retrieval process tends to be stable relative to the first iteration. As the retrieval proceeds, the iteration profile gradually approaches the true profile, and the *K_Index* box gradually shortens to below 0.5 (see Fig. 4b). Using this value as the threshold for *K_Index*, most of the Jacobian after the second



iteration does not need to be updated, and the retrieval time could be effectively reduced. However, the *K_Index* in iteration 7

shows larger outliers, indicating that the instability of the retrieval algorithm increases in the later part of the iterative process.

To reduce the impact of the Jacobian on the convergence of the algorithm, the threshold for the *K_Index* after iteration 6 is

set to 0.1 according to Fig. 4c, of which the *K_Index* box at iteration 7 is within 0.1.

## 4 Results and discussions

The simulated AERI radiation is used for retrieval to better analyze the performance of the retrieval algorithm and eliminate

the interference of other factors. An advantage of using synthetic observations is that the true atmospheric state is known,

which we can compare to the retrieval's result. Second, the errors caused by parameters in the forward model, such as the

deviation of trace gas content, the strength and temperature dependence of the water vapor continuum absorption, and the

half-widths of absorption lines, could be eliminated (Maahn et al., 2020). Third, we can control the noise level in the

synthetic measurement.

### 4.1 Retrieval process

Examples of the Fast AERIoe retrieval using the simulated spectra at various iterations are shown in Fig. 5. These profiles

represent the typical performance of each retrieval configuration at the SGP site. The entire retrieval process took 3.59 min

with 7 iterations, in which only the Jacobian of the first and second iterations were updated. The retrieved profiles converged

quickly below 1 km, with little adjustment of the temperature and humidity profile following the first iteration. For the upper

atmosphere above 1.5 km, the temperature and humidity profiles have a relatively large adjustment and gradually approach

the radiosonde profile with the iteration process. This adjustment feature of the Fast AERIoe iteration process is very similar

to AERIoe, which is determined by the information content of the AERI spectra. The information content is concentrated

near the surface, which leads to a more rapid convergence in the lowest portions of the profile. The information content of

the upper layer is less, and as such, it is necessary to reduce the value of $\gamma$ to introduce more observation information and

reduce the weight of the initial profile so that the retrieved profiles are refined to approach the radiosonde profile as the

algorithm continues to iterate.

One advantage of the optimal estimation method remains that the posterior error covariance matrix of the solution $\hat{\mathbf{S}}$

can be obtained to estimate the error of the retrieval results of each sample. The temperature and water vapor profile show a

strong correlation for the correlation coefficient matrix of $\mathbf{S}_a$ (see Fig. 6a and Fig. 6c), especially the temperature profile,

which has a high correlation coefficient above 0.6 between any two layers because of the relatively stable vertical gradient of

the temperature profile. The non-diagonal elements below 1 km in the correlation coefficient matrix of the $\hat{\mathbf{S}}$ results from



Fast AERIoe show a much lower correlation than that of $\mathbf{S}_a$ (see Fig. 6b and Fig. 6d), which means that the retrieved profiles in the lower atmosphere are dominated by the AERI spectrum. However, with the increase of height, the correlation of the area near the diagonal increases significantly. Therefore, the retrieval algorithm will rely more on the constraint of prior

information at the upper layer of PBL. The 1-σ uncertainty lines, which is the square root of the diagonal of the covariance matrices for the prior (blue shaded area) and the retrieval (black horizontal line) in Fig. 5, show that the retrieved profile has a much smaller uncertainty than the prior. Therefore, the Fast AERIoe algorithm can effectively reduce the impact of errors in the first-guess profile on the retrieval results. As the height increases, the black horizontal line segment becomes longer for the retrieval of either the temperature profile or water vapor profile, indicating that the error of the retrieval profile increases

at the upper layer.

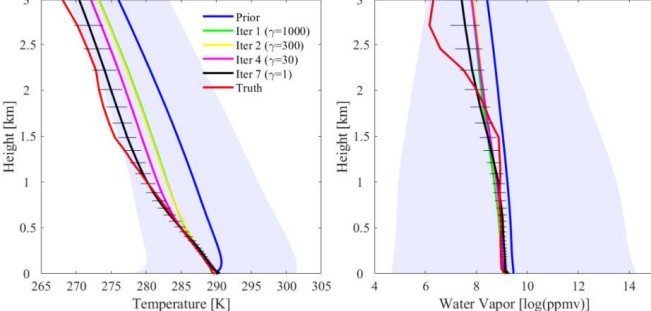

**Figure 5.** Retrieved (left) temperature and (right) WVMR at various iterations from the simulated AERI observations, where the simulated observations were computed from a radiosonde (shown in red curves) launched at the SGP site at 11:30 UTC 20 Apr 2012. The prior mean profile (blue) was used as the first guess, and the blue-shaded area illustrates the 1-σ uncertainties in the prior. The profiles at iterations 1, 2,

and 7 was shown in solid blue, yellow, purple, and black (with 1-σ error bars derived from $\hat{\mathbf{S}}$ ) lines, and the $\gamma$ were set to 1000, 300, 30 and 1 for above iterations, respectively.

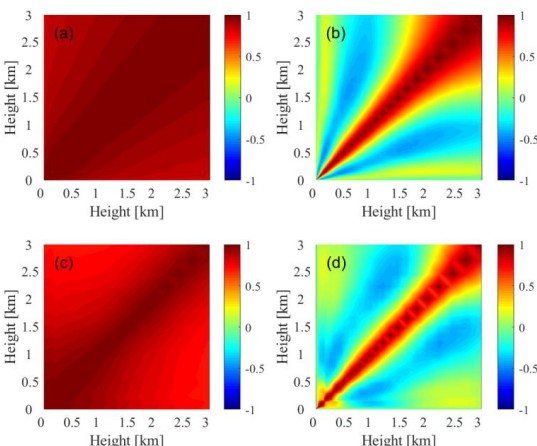

**Figure 6.** The level-to-level correlation of the prior (left) and posterior (right) for temperature–temperature (top row) and WVMR–WVMR (bottom row) at 11:30 UTC 20 Apr 2012.



**4.2 Performance**

**4.2.1 Retrieval time**

The AERIoe and Fast AERIoe algorithms were used to retrieve 826 groups of simulated AERI radiation data at SGP stations in 2012 to evaluate the retrieval performance of Fast AERIoe, with more data samples capable of ensuring representative retrieval results. The computing platform used in the retrieval process is Lenovo Aircross 510P, with the CPU Intel Core i7-7700 and the operating system Ubuntu 14.04. The average retrieval time of Fast AERIoe is 3.69 min, which is more than 50% shorter than that of AERIoe, with an average retrieval time of 8.96 min.

**Table 2.** Sample numbers of different classes according to the *K_diff* values

| Classification | *K_diff* | Sample Numbers |
|---|---|---|
| Class1 | 1 | 8 |
| Class2 | 2 | 15 |
| Class3 | 3 | 60 |
| Class4 | 4 | 193 |
| Class5 | 5 | 471 |
| Class6 | 6 | 73 |
| Class7 | 7 | 1 |

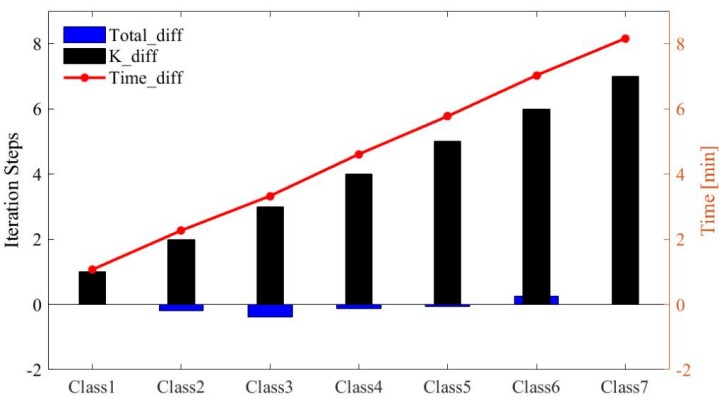

**Figure 7.** The distribution of *K_diff*, *Total_diff* and *Time_diff* with different classes.

Theoretically, the retrieval time is mainly affected by the difference in the number of iterations of the updating Jacobian and total iteration steps. The retrieval samples are divided into 7 categories (shown in Table 2) in keeping with the difference between the updating Jacobians (*K_diff* for short) of AERIoe and the Fast AERIoe. On this basis, the average retrieval time difference (*Time_diff* for short) and average total iteration step difference (*Total_diff* for short) between the two retrieval algorithms for various samples are calculated. As shown in Fig. 7, with an increase in *K_diff*, Time_ Diff also increased gradually, showing a strong positive correlation. Compared with *K_diff*, the value of *Total_diff* is very small, and its impact



on the retrieval time is also minimal, only having a slight negative and positive effect on the *Time_diff* of Calss3 and Class6. Therefore, the improvement in the retrieval speed of Fast AERIoe is mainly due to the updating process of the Jacobian.

### 4.2.2 Convergence characteristics

All samples using the AERIoe algorithm achieved convergence, with the convergence rate reaching 100%. The Fast AERIoe

algorithm has 815 groups of samples to achieve convergence, with the convergence rate reaching 98.7%, which is lower than that of AERIoe. Among the 11 sets of retrieval samples that did not achieve convergence, the *K_Index* of most of them did not change much after the $\gamma$ was dropped to 1, indicating that the subsequent iterations had little effect on the adjustment of the profiles, so the iterative profile corresponding to the very small value of the *convergence_index* could be taken as the retrieval results instead of criterion (2). Fig. 8a shows the comparison between the retrieved profiles from AERIoe using

criteria (2) and Fast AERIoe using the new convergence criteria with 11 sets of nonconverged samples. The temperature profiles obtained by the two algorithms are virtually identical, with an R-square of 0.99. For water vapor, the introduction of the new convergence criteria reduces the value of R-square but still reaches 0.84, indicating that the two datasets still have a strong correlation. The above results indicate that the method of using the *convergence_index* minimum to obtain the retrieval profiles is a reasonable and feasible method, as the Fast AERIoe algorithm cannot achieve convergence.


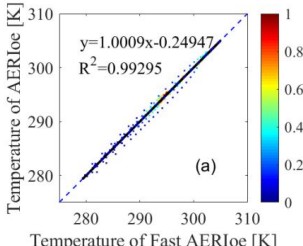 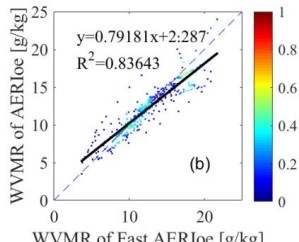

**Figure 8.** Scatter plots between the retrieval results of the nonconverged samples with AERIoe and Fast AERIoe. (a) Temperature profiles, (b) WVMR profiles.

### 4.2.3 Accuracy

Traditional methods used to evaluate the accuracy of retrieved profiles against radiosondes compute the BIAS and Root

Mean Square Error (RMSE), with the calculation formula as follows:

$$BIAS(i) = \frac{\sum_{j=1}^{M}(\mathbf{X}_{sonde}(i,j) - \mathbf{X}_{retrieval}(i,j))}{M}, \tag{11}$$




$$RMSE\,(i) = \sqrt{\frac{\sum_{j=1}^{M}(\mathbf{X}_{sonde}\,(i,j) - \mathbf{X}_{retrieval}\,(i,j))^2}{M}}\,,$$
(12)

Where $i$ and $j$ represent the serial numbers of vertical stratification and samples, respectively, with $M$ being the number of samples, $\mathbf{X}_{retrieval}$ being the retrieval result, and $\mathbf{X}_{sonde}$ is the radiosonde data.

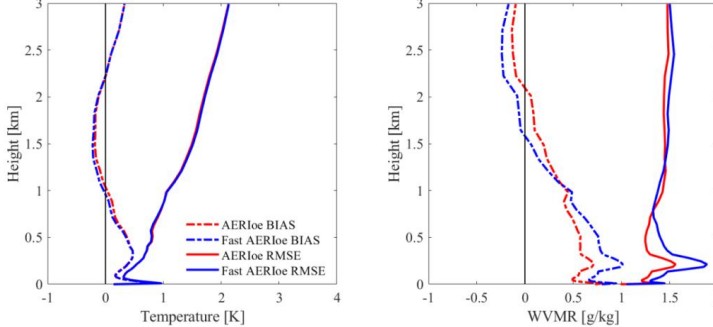

**Figure 9.** Bias (solid curves) and RMSE (dashed curves) profiles for clear-sky comparisons of the AERIoe(red curves) and Fast AERIoe (blue curves) retrievals with radiosondes. (Left) Temperature profile, (right) WVMR profiles.

The BIAS and RMSE of AERIoe and Fast AERIoe are calculated for 826 sets of samples using the above equations within the altitude range of 0-3 km, and the results are shown in Fig. 9. The temperature profile below 500 m and the water vapor profile below 1.5 km have obvious positive deviations, with the maximum deviation reaching 1.0 K and 1.0 g/kg, respectively. However, the BIAS and RMSE at the bottom are significantly reduced due to the constraint of the surface observations, indicating that the introduction of surface meteorological observation data in the observation vector has an obvious positive effect. The BIAS and RMSE of Fast AERIoe retrieved temperature profiles are in good agreement with AERIoe, with only slight differences in BIAS metrics between 500 m and 1.5 km. For the water vapor profile, the iterative strategy for the Jacobi matrix slightly changes the BIAS and RMSE profile relative to the AERIoe, with a maximum increase of 0.29 g/kg in BIAS and a maximum of 0.32 g/kg in RMSE, which is located between 0 and 0.5 km. However, the average absolute deviation is less than 0.5 g/kg in both datasets, indicating that the retrieval accuracy of Fast AERIoe is comparable to that of AERIoe overall from the index point of view.

An additional set of plots needed to be used to evaluate how well each retrieved profile can capture the vertical shapes of its true profile, as BIAS and RMSE can only describe the average accuracy of the whole dataset at each height. These Taylor diagrams show Pearson's correlation coefficient between two datasets on the y-axis and the ratio of the standard deviation on the x-axis. Retrievals that have a correlation coefficient of 1 and a standard deviation ratio (SDR) of 1 mean that the two datasets match perfectly. Fig. 10a and Fig. 10b show these plots for the clear-sky AERIoe and fast AERIoe retrievals.



For the temperature retrievals, both the Fast AERIoe and the AERIoe perform well, with 90 percent of correlation coefficients above 0.9 and the intersection of the arms infinitely close to 1. Fig. 10b shows that retrieving the water vapor structure is much more difficult with both algorithms; the spread in the correlation coefficient and SDR are much larger for water vapor than for temperature. Similar to the BIAS and RMSE profiles below 1 km in Fig. 9, Fig. 10b suggests that the AERIoe has a slightly better performance at capturing the structure of the water vapor profile than the Fast AERIoe, as the former's SDR bounds are slightly closer to 1 than the Fast AERIoe.

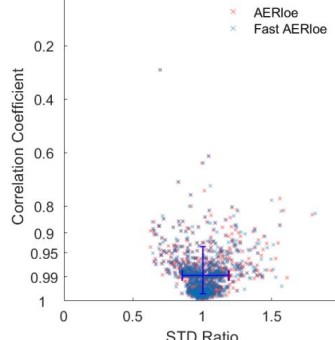 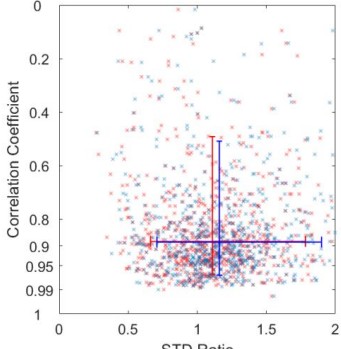

**Figure 10.** Modified Taylor plots for the retrieved clear-sky (a) temperature and (b) WVMR using the AERIoe (red symbols) and Fast AERIoe (blue symbols) datasets (826 samples). Each symbol indicates the score for an individual profile. The arms of the plotted crosses span the 10th–90th percentiles for the correlation coefficient (vertical arms) and the standard deviation ratio (horizontal arms). The intersection of the arms represents the location of the median correlation coefficient and standard deviation ratio of the given dataset.

## 4.3 Real observations

Since the clouds overhead have an significant influence on the infrared spectra, the primary problem is how to screen clear-sky samples when using the measured AERI data to retrieve the temperature and humidity profile. The interaction between clouds and infrared radiation not only interferes with the inversion of temperature and humidity profile, but also provides technical means for obtaining cloud macro parameters. Fig. 11 shows the AERI observed spectrum under cloudy and clear sky conditions. The AERI observations under the two conditions remain highly different, indicating that the AERI observed spectrum can be adopted to directly determine whether clouds or clear skies are present. To establish an accurate cloud recognition model, we adopted the cloud fraction data obtained from the all-sky image at the same site as the label, where the sample with a cloud fraction less than 30% is marked as 0, indicating clear sky, while the sample with a cloud fraction greater than 30% is marked as 1, indicating that there is cloud over head. Using the above mentioned method, the cloud fraction of the all-sky image from March to May 2010 was labeled and temporally matched with the AERI observed radiance to form a training sample set, based on which a cloud recognition model was established by training the back propagation (BP) neural network, with the final cross-validation accuracy reaching 94.3%. Compared with the recognition



method by radiosonde, the BP cloud recognition model has greatly improved the discrimination accuracy without requiring additional detection equipment. The model was used to judge 178 groups of data on October 21, 2012, with 168 groups of

380     clear sky samples screened in total.

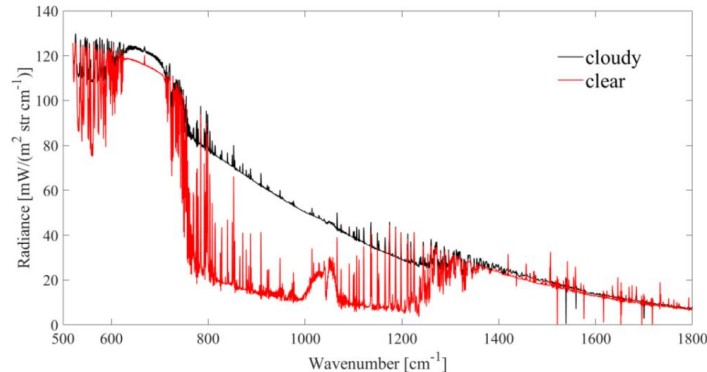

**Figure 11.** AERI observations in clear and cloudy sky conditions.

Benefiting from good retrieval accuracy and high temporal resolution, AERI instruments can be used to monitor thermodynamic temporal structures that may not be resolved by infrequent radiosonde launches. Fig. 12 shows the

385     time–height cross sections of temperature (up panel) and WVMR (bottom panel) derived from the Fast AERIoe retrievals. It can be seen from the figure that AERI resolved the temperature inversion prior to approximately 15:00 UTC, with the inversion layer height gradually rising over time. After 15:00 UTC, the temperature gradually increased, and the inversion layer gradually disappeared. From the comparisons with the radiosonde profiles shown in Fig. 13, the retrieval results of the measured radiance match the radiosonde profiles very well, especially the temperature profiles, which well reflect the

390     development and change in the inversion layer from being to not.

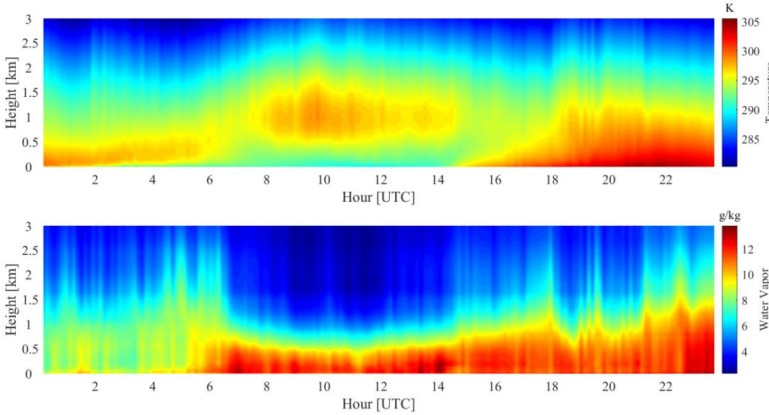

**Figure 12.** Time-height cross sections of temperature (top) and water vapor (bottom) on Oct. 21, 2012.





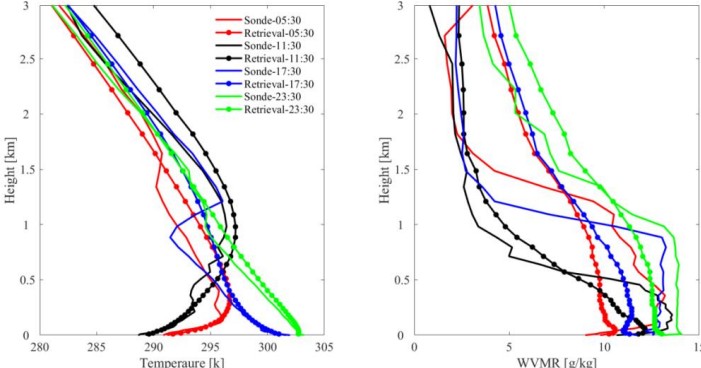

**Figure 13.** Comparisons between retrieved thermodynamic profiles and the coincident radiosonde profiles at 05:30 UTC, 11:30 UTC,
17:30 UTC and 23:30 UTC on Oct. 21, 2012. (Left) The temperature profiles, (right) the water vapor profiles.

## 5 Conclusions

The AERIoe algorithm retrieves atmospheric temperature and humidity profiles on the basis of the optimal estimation
algorithm, which can make full use of information in the infrared spectrum and give the uncertainty analysis of each retrieval.
AERIoe reduces the dependence of the retrieval process on the first-guess profile by introducing regularization parameters,
but at the same time, it also leads to more iterative steps of the retrieval algorithm, with a high calculation amount and
retrieval time of the algorithm. In this paper, a fast retrieval method called Fast AERIoe is established on the basis of AERIoe
by adaptively updating the calculation of the Jacobians. Based on the statistical comparison of the two methods (AERIoe
retrieval and Fast AERIoe retrieval) with radiosonde observations, the retrieval performance of Fast AERIoe are summarized
as follows:

    1. The retrieval speed of the Fast AERIoe is significantly improved compared with AERIoe while keeping the
parameters of the computing platform unchanged, with the average retrieval time reduced by more than 50%.

    2. Compared with AERIoe, the RMSE of Fast AERIoe retrieval is almost unchanged, illustrating that the accuracy of
Fast AERIoe is comparable to that of AERIoe. As for the convergence characteristics, all of the samples adopted AERIoe
meets the convergence criterion, while the sample adopted Fast AERIoe converged over 98% of the time. The updating
method of Jacobian in Fast AERIoe slightly reduces the convergence of the retrieval algorithm, but its convergence rate is
still acceptable.

    3. When the Fast AERIoe is adopted to measured AERI data, a cloud recognition model without additional detection
equipment is established based on the BP neural network algorithm to remove cloudy-sky cases. Compared with the
commonly used cloud recognition method by radiosonde observations, the BP cloud recognition model has greatly improved
the discrimination accuracy. It should be noted that the hyperspectra under the two weather conditions of clear sky with high

humidity and few clouds are relatively close, while the above two weather conditions are far from further distinguished when building the BP cloud recognition model, which may reduce the discriminative accuracy of the model.

A single instrument always has some defects at the vertical coverage, high vertical, temporal resolution and accuracy in obtaining the vertical distribution of atmospheric continents (Barrera-Verdejo et al., 2016). The joint detection of multiple

remote sensing devices in an optimal retrieval algorithm can overcome the shortcomings of a single device, making full use of the advantages of each detection method to achieve the purpose of enhancing their benefits. However, the increase in observation equipment will inevitably lead to more complex calculations of the forward model and Jacobian, which will lead to a significant increase in the amount of calculation and retrieval time. Therefore, it is particularly necessary to carry out research on fast retrieval in the case of joint retrieval. Apart from the influence of the Jacobian on the retrieval time, so does

the number of iterations required by the retrieval algorithm, while the number of iterations can be adjusted through the regularization parameter. The next work will focus on the application of the Fast AERIoe algorithm in joint retrieval and the selection of regularization parameters to permit the retrieval algorithm to achieve the fastest retrieval speed.

*Data availability.* The data used in the manuscript (including AERI, radiosonde, etc) are available from the ARM Data

Archive (https://adc.arm.gov/discovery/#/, accessed on 19 January 2022).

*Author contributions.* LL, BY and WH determined the main goal of this study. WH developed the approach, analyzed the data, and visualized the results of the experiments. LL prepared the paper with contributions from all co-authors. SH acquired funding and edited the paper. WL and WY prepared the various data sets. ZL and XY provided guidance on

algorithmic procedures. All the co-authors reviewed the paper.

*Competing interests.* The contact author has declared that none of the authors has any competing interests.

*Acknowledgments.* The authors thank the U. S. Department of Energy (DOE) Atmospheric Radiation Measurement (ARM)

Program for providing meteorological data online for free. The authors are deeply grateful to Atmospheric and Environmental Research (AER) Inc. for providing the LBLRTM codes online for free.

*Financial support.* This work is supported by the National Natural Science Foundation of China (Grant No. 42175154 and Grant No. 62105367) and Natural Science Foundation of Hunan Province (Grant No. 2020JJ4662).




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
