# Peer review of "Retrieval of temperature and humidity profiles from ground-based high-resolution infrared observations using an adaptive fast iterative algorithm"

_EGUsphere, 2023_

## Author Comment (AC1)

Major comments:

C1. The value of K_Index determines the iterative process of Jacobians. However, the threshold of K_Index is chosen by the distributions of the K_Index values for each iteration, which is dependent on the datasets used in the experiment. This affects the suitability of the fast retrieval algorithm. The authors should point this out. More discussions on this inadequacy of the proposed algorithm should be provided in Section 3.3.3 or in the conclusions.

R1: Thanks for the suggestion. The discussion of this issue is provided in Section 3.2.3. For your convenience, the corresponding part in our revised submission is given as follows:

It should be noted that the threshold of K_Index used in the Fast AERIoe algorithm is dependent on the datasets used in the retrieval. They are presented 'as is' and are not intended to be directly applied by the reader. We encourage readers to develop their own fast retrieval algorithms based on the atmospheric constituents they intend to retrieve.

C2. Figure 3: I am confused by the X-axis in the two panels. The authors said that IC and DFS change with K_Index are denoted with black lines, while the X-axis represents K_index is red. The illustrations of Figure 3 seems elusive to me and thus further clarification is needed in the figure caption or in the main text.

R2: We appreciate the suggestion. We admit that the logic here is indeed a bit confusing. Fig 3 and the analysis of this picture are reworked, which are given as follows:

The K_Index, ranged from 0 to 260, was obtained by multiplying the a prior profile by different scale factors. This range covers most of the K_Index during AERIoe retrieval process (see Fig. 4). The atmosphere dependent K was computed by LBLRTM with a prior profiles of different scale factors, and the SIC and DFS corresponding to the Jacobians above were calculated by equations (6) and (7), respectively. Fig. 3 shows the curve of SIC and DFS changed with K_Index. Both of them change slowly with K_Index, with the variation of SIC within 13.46% (from 13.89 to 16.05), and DFS within 4.38% (from 3.71 to to 3.88) for temperature and within 12.73% (from 1.44 to 1.65) for water vapor, which demonstrates that SIC and DFS remain almost unchanged on the condition that the value of K_Index is small. This provides an effective means to improve the retrieval speed of AERIoe by recalculating K selectively when X is not changing much or K_Index is small. This could be achieved by comparing the value of K_Index and its threshold at each iteration to determine whether K is recalculated or not.

[Figure]

Figure 3. (a) The change of SIC with K_Index. (b) The change of DFS with K_Index for temperature (unfilled circles) and water vapor (open squares), respectively.

C3. 4.2.3 Accuracy:The smoothing error cannot be ignored when retrieved profiles are compared directly to radiosondes. Thus, the radiosonde observations should be smoothed with the averaging kernel to minimize the vertical representativeness error.

R3: Thanks for pointing this problem out. We've made two modifications to Fig. 9. One is that the radiosonde observations have been smoothed with the averaging kernel **A** to reduce the vertical representativeness errors.

$$\mathbf{X}_{sonde}^{smoothed} = \mathbf{A}\left(\mathbf{X}_{sonde} - \mathbf{X}_a\right) + \mathbf{X}_a, \tag{13}$$

Another modification is the transformation of water vapor into the form of log(ppmv). This is due to the fact that the unit of **K** output by LBLRTM is log (ppmv) during the retrieval, which means that the adjustment of the iterative profile is also in the form of log (ppmv). Therefore, we believe that it is more reasonable to compare water vapor profiles of different retrieval algorithms in the form of log(ppmv).

[Figure]

**Figure 9.** Bias (solid curves) and RMSE (dashed curves) profiles for the comparisons of AERIoe(red curves) and Fast AERIoe (blue curves) retrievals with radiosondes. (Left) Temperature profile, (right) Water Vapor profiles.

C4. One subject where the manuscript lacks is the discussion on the comparison between the retrieval time and the temporal resolution of AERI spectrum. If most of the AERIoe's retrieval time exceeds the temporal resolution, then the importance of the fast retrieval algorithm will be highlighted and vice versa. Please discuss this issue.

R4: Thank you for pointing out the problem. The comparison of the retrieval time with the temporal resolution of AERI spectrum is discussed as follows:

The average retrieval time of Fast AERIoe for the 826 cases used in the study is 3.69 min, which is more than 50% shorter than that of AERIoe, with an average retrieval time of 8.96 min, which is beyond the temporal resolution (about 8 min) of AERI observations. All of the samples of AERIoe consumed more than 8 minutes, while only 10 cases exceeded the temporal resolution of AERI for Fast AERIoe algorithm. Note that the retrieval time is dependent on the computing platform and the method used to compute Jacobians and are not intended to be directly applied by the reader.

Minor comments:

C5. For the title, may be "Ground-based infrared hyperspectral retrievals of temperature and humidity profile based on Adaptive Fast Iterative Algorithm" is better.

R5: Thanks for the suggestion. The title has been modified as follows:

Ground-based infrared hyperspectral retrievals of temperature and humidity profile based on Adaptive Fast

Iterative Algorithm

C6. Line10: "due to" is usually not placed at the beginning of a sentence

R6: Thank you for pointing out the problem. The sentence in line10 is rephrased as follows:

Two methods for retrieving temperature and water vapor profiles from the Atmospheric Emitted Radiance Interferometer (AERI) observations are physical and statistical retrieval algorithms. The physical retrieval algorithm, named AERI Optimal Estimation (AERIoe), outperforms statistical retrieval algorithms in many aspects except the retrieval time, which is significantly increased due to the complex radiative transfer process.

C7. Line12: "part" -> "step"; Line15: "is" -> "was"; Line17: suggest revising to "resulting in an average retrieval time reduction from 8.96 min to 3.69 min" instead of "with the average retrieval time reduced from 8.96 min to 3.69 min"; Line41: "FTIR" -> "The FTIR instrument"; Line45: "which is more advantageous" can be revised to "which makes it more advantageous".

R7: Thanks for the constructive suggestions to improve our manuscript. We have modified the manuscript as suggested by the reviewers.

C8. Line57: this sentence should be reworked

R8: Thanks for the suggestion. The sentence in line57 is rephrased as follows:

However, the AERIprof algorithm has several significant drawbacks, such as its high sensitivity to the first-guess profile and the inability to provide uncertainty estimates for retrieval results.

---

## Author Comment (AC2)

C1. Did the authors modify the AERIoe code itself or did they develop a new code base from scratch? Please state in the Data availability if/where/how the fast AERIoe code is available. (Proprietary or open source? How does one obtain it?).

R1: Thank you for the comments. The code for the Fast AERIoe algorithm was developed by ourselves written in MATLAB language. The code for recalculating the Jacobians are not publicly available at this time but may be obtained from the authors upon reasonable request.

C2. Given that the main goal is to reduce the computation time, specifics in that regard are needed. Has the code timing been analyzed and what are the bottlenecks? I assume calculation of the Jacobian is the main bottleneck; is that the case?

R2: We appreciate the suggestion. The code timing has been analyzed as follows if we understand correctly:

The codes for the retrieval algorithm are written in MATLAB language and runs on a Lenovo Aircross 510P computer, of which the CPU is Intel Core i7-7700 and the operating system is Ubuntu 14.04. To analyze the code timing of the retrieval algorithm, the code was divided into the following sections: preparation, iteration 1, iteration 2, iteration 3,... and iteration final. The preparation section mainly consists of atmosphere construction, observation vector construction and pre-calculated variables importation. The iteration sections include the recalculation of $\mathbf{K}$ and $F(\mathbf{X})$ and the inversion using equation (1). Note that iteration 1 does not need to calculate $\mathbf{K}$ and $F(\mathbf{X})$ because the a prior profile $\mathbf{X}_a$ is fixed (mean value of the atmosphere), and the $\mathbf{K}$ and $F(\mathbf{X})$ associated with it are pre-calculated. The time consumed by each section was analyzed both for AERIoe and Fast AERIoe, results for an arbitrarily selected case are provided in Table 1. The recalculation of $F(\mathbf{X})$ and $\mathbf{K}$ consumed an immense amount of time in the retrieval process of AERIoe, and the latter is the most time consuming section. Therefore, by reducing the recalculation of $\mathbf{K}$, the retrieval time of Fast AERIoe is greatly reduced compared to AERIoe.

Table 1. List of time consumption (units: s) by the sections of AERIoe and Fast AERIoe. The sections denoted with superscript "*" indicate that $\mathbf{K}$ is not recalculated during Fast AERIoe retrieval process.

| | Sections | AERIoe | Fast AERIoe |
|---|---|---|---|
| | preparation | 0.29 | 0.22 |
| iteration 1 | inversion | 0.29 | 0.22 |
| iteration 2 | recalculation of $F(\mathbf{X})$ | 17.11 | 16.69 |
| | recalculation of $\mathbf{K}$ | 68.76 | 70.27 |
| | inversion | 0.31 | 0.27 |
| iteration 3 | recalculation of $F(\mathbf{X})$ | 17.18 | 17.04 |
| | recalculation of $\mathbf{K}$ | 70.55 | 0.00 |
| | inversion | 0.22 | 0.22 |
| iteration 4 | recalculation of $F(\mathbf{X})$ | 17.71 | 16.36 |
| | recalculation of $\mathbf{K}^*$ | 70.07 | 0.00 |
| | inversion | 0.25 | 0.21 |
| iteration 5 | recalculation of $F(\mathbf{X})$ | 16.97 | 17.38 |
| | recalculation of $\mathbf{K}^*$ | 68.93 | 0.00 |
| | inversion | 0.21 | 0.25 |
| iteration 6 | recalculation of $F(\mathbf{X})$ | 16.08 | 15.08 |
| | recalculation of $\mathbf{K}^*$ | 68.23 | 0.00 |
| | inversion | 0.24 | 0.24 |
| iteration final | recalculation of $F(\mathbf{X})$ | 15.91 | 18.45 |

| | | |
|---|---|---|
| recalculation of $\mathbf{K}^*$ | 68.11 | 0.00 |
| inversion | 0.28 | 0.23 |

C3. Please begin with a sentence that more clearly gives the background - something like: "Two methods for retrieving … are physical and statistical retrieval algorithms …"

R3: Thanks for the constructive suggestions to improve our manuscript. The corresponding part in our revised submission is given as follows:

Two methods for retrieving temperature and water vapor profiles from the Atmospheric Emitted Radiance Interferometer (AERI) observations are physical and statistical retrieval algorithms. The physical retrieval algorithm, named AERI Optimal Estimation (AERIoe), outperforms statistical retrieval algorithms in many aspects except the retrieval time, which is significantly increased due to the complex radiative transfer process.

C4. Line 12: Begins with "Further analysis showed…" but no analysis has yet been discussed. What changes were made to the Jacobians and why was that expected to speed up performance (but didn't)?

R4: Thank you for pointing out this problem. We have modified the problem in our revised submission. The corresponding revised part in our revised submission is given as follows:

Analysis of the change of AERI observations' information content with Jacobians revealed that the performance of AERIoe algorithm had little dependence on Jacobians. Thus, the Jacobian matrix could remain unchanged when the variation of iterative profiles is small in the retrieval process. This significantly reduces the amount of computation and thus increases the retrieval speed of AERIoe.

C5. The time estimates are not useful without knowing what type of computing platform was used. Perhaps just give the percent improvement. Also, are 3 significant figures warranted here?

R5: Thank you for the suggestion. We have modified the problem in our revised submission. The corresponding revised part in our revised submission is given as follows:

The retrieval speed was significantly improved compared with the original AERIoe algorithm under the condition that the parameters of the computing platform remain unchanged, resulting in an average retrieval time reduction by 58.82%.

C6. What is meant by "certain impact"? What is meant by "to some extent"? Why not state the convergence rate of the traditional algorithm?

R6: Thank you for pointing out the problems. We have modified them in our revised submission. The corresponding revised part in our revised submission is given as follows:

Results based on synthetic observations revealed that the fast retrieval algorithm reached an acceptable convergence rate of 98%, which is slightly lower than the 99.88% convergence rate of AERIoe for the 826 cases used in this study.

C7. The authors say that "The retrieval accuracy of the fast retrieval model is equivalent to that of the traditional algorithm." However, on lines 346-348 differences indicate that the accuracies are not equivalent.

R7: Thank you for pointing this problem out. We have modified the sentence as "The retrieval results of the fast retrieval model are comparable to that of AERIoe." Reasons for this modification are discussed in detail in the response to comments on lines 346 - 348.

C8. How is the convergence criteria adjusted to give reliable retrieval results? It was previously stated that the results were equally accurate. Do you mean they are equally accurate when they both converge?

R8: Thank you for the comments. The sentence 'However, reliable retrieval results can still be obtained by adjusting the convergence criteria.' has been removed.

C9. Line 115: If the authors are using X0 = Xa, they should replace X0 with Xa in Eqn (1) so it is consistent with Turner and Lohnert. If not, they should explain this change.

R9: Thanks for the suggestion. We have replaced $\mathbf{X}_0$ with $\mathbf{X}_a$ in Eqn (1).

C10. Line 116 says "Y is the observed radiance vector, F(X) is the AERI observed spectrum…" Is it rather that Y is the observed radiance vector (from the observed AERI spectrum) and F(X) is the estimate of Y from the forward model calculation? It would also be helpful to define that the background refers to the a priori atmospheric state, if that is the case.

R10: Thanks for the suggestion. We have replaced '$F(\mathbf{X})$ is the AERI observed spectrum' with '$F(\mathbf{X})$ is the computed radiance for $\mathbf{X}$'. The sentence '$\mathbf{S}_a$ it the background covariance matrix' has been changed to '$\mathbf{S}_a$ is the a priori covariance matrix'.

C11. Eqn. 1: I'm curious why this formulation is used instead of the Levenberg-Marquardt formulation (Rodgers 2000, Eqn 5.36). How is the behavior the same or different? Carissimo et al. 2005 state that their method is almost equivalent to Levenberg-Marqardt. In Levenberg-Marquardt, increasing gamma decreases the step size and makes the retrieval weighted more toward steepest descent. How is the formulation here the same or different?

R1: Thank you for the comments, we have put our thinking caps on this question and the discussion regarding it is presented as follows:

The iterative equation for AERIoe is as follows

$$\mathbf{X}_{n+1} = \mathbf{X}_a + \left(\mathbf{K}_n^T \mathbf{S}_e^{-1} \mathbf{K}_n + \gamma \mathbf{S}_a^{-1}\right)^{-1} \mathbf{K}_n^T \mathbf{S}_e^{-1} \times \left(\mathbf{Y}^m - F(\mathbf{X}_n) + \mathbf{K}_n(\mathbf{X}_n - \mathbf{X}_a)\right) \tag{1}$$

The equation for the Levenberg-Marquardt method is given as follows (Rodgers, 2000)

$$\mathbf{X}_{n+1} = \mathbf{X}_n + \left(\mathbf{K}_n^T \mathbf{S}_e^{-1} \mathbf{K}_n + (1+\gamma)\mathbf{S}_a^{-1}\right)^{-1} \times \left(\mathbf{K}_n^T \mathbf{S}_e^{-1}\left(\mathbf{Y}^m - F(\mathbf{X}_n)\right) - \mathbf{S}_a^{-1}(\mathbf{X}_n - \mathbf{X}_a)\right) \tag{2}$$

We believe that the comparison of the two methods can be analyzed from two aspects: the retrieval process and the retrieval results.

(1) In terms of retrieval process, Carissimo et al. (2005) state that gama in Eqn (1) dumps the width of the step between two consecutive iterates and leads its direction toward the steepest descent of the cost function. The gama in Eqn (2) is chosen at each step to reduce the cost function and also tends to the steepest decent of cost function. Therefore, the role of gama in Levenberg-Marquardt method is equivalent to that of AERIoe.

However, the values of gama in the two formulas are quite different, as the profiles in Eqn (1) are retrieved by adjusting the a prior profile $\mathbf{X}_a$, while the profiles in Eqn (2) are iterated by adjusting the iterative profile $\mathbf{X}_n$. For example, in the work of Foth and Pospichal (2017) the initial value of gama is 2, and increases by a factor of 10 if the cost function $\mathbf{J}$ has increased and reduces by a factor of 2 if $\mathbf{J}$ has decreased. Therefore, the value of gama in Levenberg-Marquardt method shows a significant difference from AERIoe [$\gamma = 1000, 300, 100, 30, 10, 3, 1,...$].

(2) In terms of retrieval results, gama in Equation (1) needs to be set to 1 in the final step to eliminate regularization errors in the solutions, which gives

$$\mathbf{X}_{n+1} = \mathbf{X}_a + \left(\mathbf{K}_n^T \mathbf{S}_e^{-1} \mathbf{K}_n + \mathbf{S}_a^{-1}\right)^{-1} \mathbf{K}_n^T \mathbf{S}_e^{-1} \times \left(\mathbf{Y}^m - F(\mathbf{X}_n) + \mathbf{K}_n(\mathbf{X}_n - \mathbf{X}_a)\right) \tag{3}$$

It can be seen that $\left(\mathbf{K}_n^T \mathbf{S}_e^{-1} \mathbf{K}_n + \mathbf{S}_a^{-1}\right)^{-1} \cdot \mathbf{K}_n^T \mathbf{S}_e^{-1} \mathbf{K}_n$ is equivalent to $I - \left(\mathbf{K}_n^T \mathbf{S}_e^{-1} \mathbf{K}_n + \mathbf{S}_a^{-1}\right)^{-1} \cdot \mathbf{S}_a^{-1}$, thus Eqn (3) can be written as

$$\mathbf{X}_{n+1} = \mathbf{X}_a + \left(\mathbf{K}_n^T \mathbf{S}_e^{-1} \mathbf{K}_n + \mathbf{S}_a^{-1}\right)^{-1} \mathbf{K}_n^T \mathbf{S}_e^{-1}\left(\mathbf{Y}^m - F(\mathbf{X}_n)\right) + \left(I - \left(\mathbf{K}_n^T \mathbf{S}_e^{-1} \mathbf{K}_n + \mathbf{S}_a^{-1}\right)^{-1} \mathbf{S}_a^{-1}\right)(\mathbf{X}_n - \mathbf{X}_a) \tag{4}$$

So the solution becomes

$$\mathbf{X}_{n+1} = \mathbf{X}_n + \left(\mathbf{K}_n^T \mathbf{S}_e^{-1} \mathbf{K}_n + \mathbf{S}_a^{-1}\right)^{-1} \times \left(\mathbf{K}_n^T \mathbf{S}_e^{-1}\left(\mathbf{Y}^m - F(\mathbf{X}_n)\right) - \mathbf{S}_a^{-1}\left(\mathbf{X}_n - \mathbf{X}_a\right)\right)$$  (5)

Eqn (5) is a special case of Eqn (2) when gama in Eqn (2) is chosen to be 0. Therefore, the Levenberg-Marquardt is a combination of a Gauss-Newton (without gama) and steepest descent minimization technique and equivalent to AERIoe.

References:
Rodgers, C. D.: Inverse methods for atmospheric sounding: theory and practice, World scientific, 119-120 pp., ISBN9814498688, 2000.
Foth, A. and Pospichal, B.: Optimal estimation of water vapour profiles using a combination of Raman lidar and microwave radiometer, Atmos. Meas. Tech., 10, 3325-3344, https://doi.org/10.5194/amt-10-3325-2017, 2017.

C12. Figure 1: This figure needs improvement and explanation. E.g. please define "iterative observations" and "iterative profiles" in the caption. Use of the symbol "Sa" is inconsistent with use of "Jacobians" instead of "K". K_Index has not yet been defined.

R12: Thanks for the suggestion. We have changed "iterative observations" to "$F(\mathbf{X})$" and "iterative profiles" to "$\mathbf{X}_n$". In order to be consistent with "$\mathbf{S}_a$", the symbol "$\mathbf{K}$" was used instead of "Jacobians". A symbol "compute monitoring index" was used instead of "K_Index" and defined it in the caption as follows:

The monitoring index reflects the variations of $\mathbf{X}_n$.

C13. Line 118: I don't think n is the number of iterations, but rather the iteration number.

R13: Thanks for the suggestion. We have changed 'the number of iterations' to 'the iteration number'.

C14. Line 120: The description of how gamma is used is not clear.

R14: Thanks for the suggestion. The description of how gamma is used is given as follows:

AERIoe begins with scalar $\gamma$ of large values is to stabilize the retrieval and ends with a unity $\gamma$ to add more information from AERI observations as $n$ increases. This approach allows the AERIoe algorithm to overcome a poor first guess and achieve a results that have the most information from AERI observation, the detailed description of how $\gamma$ is used can be found in Turner and Löhnert (2014).

C15. Line 122: Remove "progress".

R15: Thanks for the suggestion. The word 'progress' in Line 122 has been removed.

C16. Line 122: Please change "is not allowed to converge until…" to "Iterations are continued until…" if that is what is meant here.

R16: Thanks for the suggestion. This sentence was modified as "Iterations are continued until $\gamma$ decreases to 1 and the following convergence criterion is satisfied."

C17. Line 124: Use consistent symbols. You have superscript n sometimes and subscript n other times.

R17: Thanks for the suggestion. We have changed all of superscript n into subscript n.

C18. Line 214-215: It is not true that "what affects IC and DFS lies only in gamma and Jacobian". In fact, when gamma = 1, IC and DFS are determined by Se and Sa, with the purpose of the Jacobian being to transform Se into the state space for Sa, so that they have the same units and size (rows and columns). I think what you mean is that IC and DFS only change with iteration due to changes in gamma and the Jacobian. (But see below).

R18: We appreciate the suggestion. The sentence in line 214-215 is rephrased as follows:

It can be seen from equations (6) and (7) that IC and DFS are determined by $\mathbf{S}_e$, $\mathbf{S}_a$, $\mathbf{K}$ and $\gamma$. However, $\mathbf{S}_a$ and $\mathbf{S}_e$ remain unchanged during retrieval, which makes IC and DFS only change with iteration due to variations in $\gamma$ and $\mathbf{K}$. As $\gamma$ drops to 1 at the final iteration, the values of IC and DFS are only dependent on $\mathbf{K}$.

C19. Lines 214 - 232: I don't understand the logic here. On line 224, it is strange to say that gamma changes with the adjustment of the profile, since gamma is prescribed. Figure 3 is confusing. The x-axis goes in the reverse direction as the retrieval proceeds, the figure caption description seems to be wrong (red is actually K_index), and it isn't stated where K_index starts and ends (starts at the high end, ends at the low end?). It is not surprising that the DFS and IC increase as gamma drops to 1, since gamma weights the retrieval away from the observation and toward the first-guess, which presumably has no information content at all. It is also not surprising that there is not much change in DFS and IC with the Jacobian, since, as stated previously, the purpose of the Jacobian here is to transform Se onto the dimensions of Sa. I don't see how this shows that the change of the Jacobian has less influence on the retrieval ability than gamma. Gamma is not supposed to influence the retrieval ability, but only the retrieval stability. That is why iterations are continued until gamma is 1, whereupon the retrieval equation is equivalent to the Gauss-Newton formulation and the maximum information content is used. In fact, I don't see the point of this paragraph or figure at all. The authors could simply state that if X is not changing much, as evidenced by the K_Index, then the Jacobian is probably not changing much either, and therefore does not need to be recomputed. (Note, however, that this is not necessarily true, and they need to show that it is an ok approximation).

R19: We appreciate the suggestion. We admit that the logic here is indeed a bit confusing. Fig 3 and the analysis of this picture are reworked, which are given as follows:

The K_Index, ranged from 0 to 260, was obtained by multiplying the a prior profile by different scale factors. This range covers most of the K_Index during AERIoe retrieval process (see Fig. 4). The atmosphere dependent $\mathbf{K}$ was computed by LBLRTM with a prior profiles multiplied by different scale factors, and the SIC and DFS corresponding to the Jacobians above were calculated by equations (6) and (7), respectively. Fig. 3 shows the curve of SIC and DFS changed with K_Index. Both of them change slowly with K_Index, with the variation of SIC within 13.46% (from 13.89 to 16.05), and DFS within 4.38% (from 3.71 to to 3.88) for temperature and within 12.73% (from 1.44 to 1.65) for water vapor, which demonstrates that SIC and DFS remain almost unchanged as K_Index increases on the condition that the value of K_Index is small. This provides an effective means to improve the retrieval speed of AERIoe by recalculating $\mathbf{K}$ selectively when $\mathbf{X}$ is not changing much or K_Index is small. This could be achieved by comparing the value of K_Index and its threshold at each iteration to determine whether $\mathbf{K}$ is recalculated or not.

[Figure]

Figure 3. (a) The change of SIC with K_Index. (b) The change of DFS with K_Index for temperature (unfilled circles) and water vapor (open squares), respectively.

C20. Turner and Lohnert state that "Future versions of AERIoe will use the Carissimo et al. (2005) approach in order to more efficiently converge and reduce computational time." Did the authors explore that approach, and how might that change their analysis?

R20: Thank you for the comments. In fact, the first approach we studied to reduce the retrieval time happens to be the L-curve method recommended by Turner and Lohnert (2014). In our study the codes in Regularization Tools developed by Per Christian Hansen were used to calculate the L-curve and locate the corner, and the retrieval results using gama from the L-curve method did not show any superiority over the method used in AERIoe. The reason lies in that the gama obtained from L-curve method does not gradually decrease with iterations in the retrieval process. In order to figure out the impact of gama on the iterations of AERIoe, we find that the change of **K** is negligible during most of the retrieval process, which inspired this study to reduce the retrieval time by recalculating **K** selectively.

C21. Sections 3.1 and 3.2. are unclear. The description of the retrieval forms is confusing. Is the state vector comprised of the temperature and log of water vapor on the 37 atmospheric layers? Why isn't it parametrized, given that there are far fewer degrees of freedom? It continues to be difficult to tell what is new here and what is the same as previous work. Please avoid repeating details where you could reference the previous work. For example, you could say, "The forward model is the same as that described by Turner and Lohnert, except as follows…" Was LBLRTM used here to apply the spectral response function, in contrast to the previous work? It is stated that LBLRTM can be used to calculate the Jacobian. Was it used for this purpose? Again, is this a departure from previous work?

R21: Thanks for the suggestion. We have removed Sections 3.1 and 3.2. from the manuscript and added descriptions of Fast AERIoe configurations that differ from AERIoe in the paragraph between line128 and line136. The new paragraph is reorganized as follows:

Note that **K** depends on **X** used for estimating the Jacobian, which means that **K** must be recomputed for every iteration step. The updating of the Jacobians in the retrieval process requires the calculation of the optical thickness or radiance (intensity) with respect to different atmospheric constituents at each height, which might be computationally expensive depending on the lengths of **X** and $\mathbf{Y}_m$ (Maahn et al., 2020). Owing to the constraints of $\gamma$, the decrease of the difference

between simulated and observed radiation is not very much in the adjustment of individual iterations to the retrieval profile. At this time, the change in the Jacobian calculated as per the iteration profile is negligible. Backed by the above analysis, a fast iterative algorithm called Fast AERIoe is proposed on the basis of the AERIoe algorithm. The flowchart of Fast AERIoe is shown in Fig. 1, most of the configurations are consistent with AERIoe described by Turner and Löhnert (2014), except some modifications highlighted as follows:

a. atmospheric configurations: The height grid of $\mathbf{X}$ is consistent with AERIoe, but the maximum retrieval height is limited to 3 km. This is done because the variations of K above 3 km is negligible due to the fact that most of the information content in AERI spectrum lies in the lowest 2 km of the atmosphere for temperature and water vapor profiles (Turner and Löhnert, 2014). The cloud properties were excluded from the state vector $\mathbf{X}$, which is beyond the scope of this study. The corresponding priori profile $\mathbf{X}_a$ and the a priori represented by the Sa covariance matrices are modified to be consistent with $\mathbf{X}$.

b. observational vector $\mathbf{Y}_m$: Spectral regions that sensitive to cloud properties were removed from the observational vector $\mathbf{Y}_m$ to be consistent with the state vector $\mathbf{X}$. Furthermore, additional observations including surface temperature and water vapor were incorporated into the observation vector, details are described by Turner et al. (2019).

c. Jacobian matrix K: K is derived from the line-by-line radiative transfer model (LBLRTM; Clough et al. 1992), which is the same as AERIoe except the version (12.8 instead of 12.1). Another modification is that K is not recomputed to improve the retrieval speed of the algorithm when the variations of the iterative profile $\mathbf{X}_n$ are small.

C22. Line 176: Please remove the statement that LBLRTM is the most accurate forward model or provide a reference for it.

R22: Thanks for the suggestion. The statement that LBLRTM is the most accurate forward model has been removed.

C23. Line 196-197: Please rephrase this: "…determined whether updating or not by monitoring the indicators that can reflect the changes of Jacobian in the iterative process".

R23: Thanks for the suggestion. The sentence in line 196-197 has been modified as follows:

Adaptive updating of $\mathbf{K}$ is the key to reduce the calculation amount of the AERIoe algorithm. The Jacobians are dependent on the atmospheric constituents, which means that $\mathbf{K}$ must be recalculated for every iteration step. The question arises as to under what circumstances $\mathbf{K}$ does not need to be recalculated. Therefore, the dependence of the retrieval capability on Jacobians must be analyzed and the indicators that reflect the changes of Jacobians should be figured out to determine whether $\mathbf{K}$ recalculated or not.

C24. Line 344: Please change, "with only slight differences in BIAS metrics between 500 m and 1.5 km" to include a quantitative value, such as "with differences within x% to y%"

R24: Thanks for the suggestion. The sentence "with only slight differences in BIAS metrics between 500 m and 1.5 km" was changed to "with maximum differences within 0.06 K in BIAS metrics between 500 m and 1.5 km."

C25. Lines 346 - 348: It seems like "a maximum increase of 0.29 g/kg in BIAS and a maximum of 0.32 g/kg in RMSE" are significant.   The bias increase appears to be up to about 40% (0.7 to 0.9), and the RMS increase up to ~12%. This does not seem to me to be comparable retrieval accuracy. Please clarify.

R25: Thanks for pointing this problem out. We've made two modifications to Fig. 9. One is that the radiosonde observations have been smoothed with the averaging kernel $\mathbf{A}$, as suggested by Reviewer #1, to reduce the vertical representativeness errors.

$$\mathbf{X}_{sonde}^{smoothed} = \mathbf{A}\left(\mathbf{X}_{sonde} - \mathbf{X}_a\right) + \mathbf{X}_a , \tag{13}$$

Another modification is the transformation of water vapor into the form of log(ppmv). This is due to

the fact that the unit of **K** output by LBLRTM is log (ppmv) during the retrieval, which means that the adjustment of the iterative profile is also in the form of log (ppmv). Therefore, we believe that it is more reasonable to compare water vapor profiles of different retrieval algorithms in the form of log(ppmv).

[Figure]

**Figure 9.** Bias (solid curves) and RMSE (dashed curves) profiles for the comparisons of AERIoe(red curves) and Fast AERIoe (blue curves) retrievals with radiosondes. (Left) Temperature profile, (right) Water Vapor profiles.

The Fast AERIoe retrieved temperature profiles shows a negative deviation of 0.05K between 1.0 km and 1.5 km and a maximum increase of RMSE up to 13.3% (from 0.60 K to 0.68 K) above 1.0 km when compared with AERIoe. For the water vapor profile, the BIAS and RMSE profiles of Fast AERIoe are in good agreement with AERIoe, except for a maximum increase of BIAS up to 25% (from 0.12 log(ppmv) to 0.15 (ppmv)) bellow 1.0 km. Therefore, we agree with the reviewers' comments that Fast AERIoe does not reach a comparable retrieval accuracy. However, we believe that the retrieved profiles rather than the retrieval accuracy of the two algorithms are comparable. This is because the increment of BIAS (within 0.03 log(ppmv)) is much smaller than the value (roughly on the order of 5-10 log(ppmv)) of the retrieved water vapor profile itself, and this increment had little impact on the retrieval results of AERIoe. The comparison of the profiles retrieved by the two algorithms can be demonstrated more clearly by the modified Taylor plots (see Fig. 10). Most of the blue and red symbols '×' in Fig. 10 , which indicate the scores for the individual profiles of the two algorithms, are closed to each other. Therefore, the retrieval results of the AERIoe and Fast AERIoe algorithms are comparable both for temperature and water vapor profiles.

To be more precisely, we have rephrased all the sentence like "the retrieval accuracy of Fast AERIoe is comparable to that of AERIoe" in the manuscript to "the retrieved profiles of Fast AERIoe is comparable to that of AERIoe.

[Figure]

**Figure 10.** Modified Taylor plots showing the correlation coefficient and standard deviation ratio between the smoothed radiosondes and the retrieved (a) temperature and (b) WVMR profiles using AERIoe (red symbols) and Fast AERIoe (blue symbols). There are 826 cases from the SGP site within 2012. Each symbol indicates the score for an individual profile. The arms of the plotted crosses span the 10th–90th percentiles for the correlation coefficient (vertical arms) and the standard deviation ratio (horizontal arms).

C26. Line 350: More detail is needed about how you calculated "Pearson's correlation coefficient between two datasets on the y-axis and the ratio of the standard deviation on the x-axis", and the caption of Fig. 10 needs to be improved.

R26: The details to calculate the variables in Fig. 10 are given as follows:

Each retrieval/sonde pair is used to derive the correlation coefficient (r) from Eq. (14) and the ratio of the standard deviations from Eq.(15), both are used by Turner and Löhnert (2014).

$$r = \frac{\frac{1}{N}\sum_{z=0}^{z=h}[s(z)-\bar{s}][a(z)-\bar{a}]}{\sigma_s \sigma_a}, \tag{14}$$

$$SDR = \sigma_a / \sigma_s, \tag{15}$$

Within the equations, $s(z)$ and $a(z)$ are defined as the radiosonde observations and retrieved profiles between 0 and 3 km, $(\bar{s}, \bar{a})$ and $(\sigma_s, \sigma_a)$ are the mean values and standard deviations at the same height range.

We remove the sentence 'The intersection of the arms represents the location of the median correlation coefficient and standard deviation ratio of the datasets.' in the caption of Fig.10. The caption of Fig. 10 is modified as follows:

Modified Taylor plots showing the correlation coefficient and standard deviation ratio between the smoothed radiosondes and the retrieved (a) temperature and (b) WVMR profiles using AERIoe (red symbols) and Fast AERIoe (blue symbols). There are 826 cases from the SGP site within 2012. Each symbol indicates the score for an individual profile. The arms of the plotted crosses span the 10th – 90th percentiles for the correlation coefficient (vertical arms) and the standard deviation ratio (horizontal arms).

References:

C27. For use of FTIR viewing solar spectra, you could also reference the work of Kimberly Strong's group; e.g.: https://amt.copernicus.org/articles/7/1547/2014/
For retrievals from AERI, please add a reference Rowe et al. 2006, which used constrained linear inversion to retrieve temperature profiles from an AERI instrument: Rowe, P.M., Walden, V.P. and

Warren, S.G., 2006. Measurements of the foreign-broadened continuum of water vapor in the 6.3 μm band at− 30° C. Applied optics, 45(18), pp.4366-4382.

R27: Thank you for the suggestion. We have read the references recommended by the reviewers carefully. The two references have been added into the revised manuscript.

C28. English grammar and clarity:   If possible, please have a native English speaker edit your paper throughout, including use of "the" in English, which is very challenging to get right. Examples:

Line 25 and throughout: When you are talking about something in general, omit the word "the" and change the noun to the plural.   Examples: change "the observation network" to "observation networks". Change "the convective scale numerical weather prediction system"  to "convective scale numerical weather prediction systems".   Change "the radiosonde profiles" to "radiosonde profiles".   Only use "the" if you are talking about a specific thing, and if you have made it clear which one you are talking about. For example, on line 73, add "the" before "ARM program" since it is clear which program you are talking about (ARM).

R28: Thank you for the suggestion. We have modified the issues identified by the authors in the revised manuscript. Moreover, we have carefully polished our paper to improve the English writing. If there are still writing problems with our manuscript, please let us know, and we will try our best to revise our article.

C29. Line 26: Please define all acronyms (e.g. NWP model)

R29: Thank you for the suggestion. We have checked our manuscript carefully and modified the similar problems in our revised submission.

C30. Line 32: Change "shows" to "has" or "demonstrates". Line 57: remove "to boot".

R30: Thank you for the suggestion. We have modified the problems in our revised submission.

---

## Author Response (AR1)

**Response to EGUSPHERE-2023-637 Review**

WeiHuang , Lei Liu , BinYang , Shuai Hu , Wanying Yang , Zhenfeng Li , Wantong Li and XiaofanYang

Dear editor,

We thank you very much for processing our manuscript entitled "Retrieval of temperature and humidity profiles from ground-based high-resolution infrared observations using an adaptive fast iterative algorithm" (ID: EGUSPHERE-2023-637). We also thank the referees for their comments, which are all valuable and very helpful for improving our work. We have studied comments carefully and have made extensive modifications which are marked in a smaller font size in our revised submission manuscript. We have addressed the issues pointed out by the referees, to which we respond in detail below. We hope that the referees will find our responses satisfactory, and we are willing to further revise the manuscript regarding any additional suggestions that the referees may have. Please find below the referees' comments in light blue with our responses after each comment.

**Referee #1**

Dear Referee:

We would like to express our sincere appreciation for your careful reading and invaluable comments to improve this paper. We have studied carefully on these detailed comments. They are helpful to improve our manuscript. After deep consideration of your valuable comments, we substantially modify the manuscript. Corresponding changes are marked in a smaller font size in the modified manuscript. Point-by-point responses to your comments are seriously completed for your consideration. If there are still severe issues with our manuscript, please let us know, and we will try our best to modify our article. Thanks for your time!

Major comments:

C1.The value of K_Index determines the iterative process of Jacobians. However, the threshold of K_Index is chosen by the distributions of the K_Index values for each iteration, which is dependent on the datasets used in the experiment. This affects the suitability of the fast retrieval algorithm. The authors should point this out. More discussions on this inadequacy of the proposed algorithm should be provided in Section 3.3.3 or in the conclusions.

R1: Thanks for the suggestion. The value of K_Index is dependent on the height range of atmospheric profiles and the atmospheric constituents that intended to retrieve. They are presented 'as is' and are not intended to be directly applied to the retrieval of different atmospheric profiles or instruments. The discussion of this issue is provided in Section 3.2.3 of the revised manuscript.

**For your convenience, the corresponding part in our revised submission is given as follows:**

It should be noted that the threshold of K_Index used in the Fast AERIoe algorithm is dependent on the datasets used in the retrieval. They are presented 'as is' and are not intended to be directly applied by the reader. We encourage readers to develop their own indicator to reduce the recalculation of Jacobians based on the atmospheric constituents they intend to retrieve.

C2.Figure 3: I am confused by the X-axis in the two panels. The authors said that IC and DFS change with K_Index are denoted with black lines, while the X-axis represents K_index is red. The illustrations of Figure 3 seems elusive to me and thus further clarification is needed in the figure

caption or in the main text.

R2: We appreciate the suggestion. We admit that the logic here is indeed a bit confusing. Some modifications have been made to Fig 3 and the analysis of this picture. The indicator "IC" was changed to Shannon Information Content (SIC) to be consistent with the work described by Turner and Löhnert (2014). The change of K_Index with factor γ was removed since it may lead to some confusion and occasional misunderstanding. Fig 3 and the analysis of this figure were reworked. Thanks for the suggestion again. We added the detailed discussion in the revised manuscript.

**For your convenience, the corresponding revised part in our revised submission is given as follows:**

[Figure]

Figure 2. (a) The change of *SIC* with *K_Index*. (b) The change of *DFS* with *K_Index* for temperature (unfilled circles) and water vapor (open squares), respectively.

The values of *K_Index*, which covers most of the *K_Index* during AERIoe retrieval process (ranged from 0 to 260, see Fig. 3), were obtained by multiplying the a prior profile by different scale factors. The atmosphere dependent **K** were computed by LBLRTM with the prior profiles above, and *SIC* and *DFS* were calculated using equations (3) and (4) with different Jacobians, respectively. Fig. 2 shows the curve of *SIC* and *DFS* changed with *K_Index*. Both of them change slowly with *K_Index*, with the variation of *SIC* within 13.46% (from 13.89 to 16.05), and *DFS* within 4.38% (from 3.71 to to 3.88) for temperature and within 12.73% (from 1.44 to 1.65) for water vapor, which demonstrates that *SIC* and *DFS* remain almost unchanged as *K_Index* increases on the condition that the value of *K_Index* is small. This provides an effective means to improve the retrieval speed of AERIoe by recalculating **K** selectively when **X** is not changing much or *K_Index* is small. This could be achieved by comparing the value of *K_Index* and its threshold at each iteration to determine whether **K** is recalculated or not.

C3. 4.2.3 Accuracy:The smoothing error cannot be ignored when retrieved profiles are compared directly to radiosondes. Thus, the radiosonde observations should be smoothed with the averaging kernel to minimize the vertical representativeness error.

R3: Thanks for pointing this problem out. We've made two modifications to Fig.9. One is that the radiosonde observations have been smoothed with the averaging kernel **A** to reduce the vertical representativeness errors.

$$\mathbf{X}_{sonde}^{smoothed} = \mathbf{A}(\mathbf{X}_{sonde} - \mathbf{X}_a) + \mathbf{X}_a$$

Another modification is the transformation of water vapor into the form of log(ppmv). This is due to the fact that the unit of **K** output by LBLRTM is log (ppmv) during the retrieval, which means that

the adjustment of the iterative profile is also in the form of log (ppmv). Therefore, we believe that it is more reasonable to compare water vapor profiles of different retrieval algorithms in the form of log(ppmv).

**For your convenience, the corresponding revised part in our revised submission is given as follows:**

... $\mathbf{X}_{sonde}^{smooth}$ is radiosonde observations which are smoothed with the averaging kernel A by the following multiplication to reduce the vertical representativeness errors

$$\mathbf{X}_{sonde}^{smoothed} = \mathbf{A}\left(\mathbf{X}_{sonde} - \mathbf{X}_a\right) + \mathbf{X}_a, \tag{10}$$

The BIAS and RMSE of AERIoe and Fast AERIoe are calculated for 826 sets of samples using the above equations within the altitude range of 0-3 km, and the results are shown in Fig. 8.

[Figure]

**Figure 8.** Bias (solid curves) and RMSE (dashed curves) profiles for the comparisons of AERIoe(red curves) and Fast AERIoe (blue curves) retrievals with radiosondes. (Left) Temperature profile, (right) Water Vapor profiles.

C4. One subject where the manuscript lacks is the discussion on the comparison between the retrieval time and the temporal resolution of AERI spectrum. If most of the AERIoe's retrieval time exceeds the temporal resolution, then the importance of the fast retrieval algorithm will be highlighted and vice versa. Please discuss this issue.

R4: Thank you for pointing out the problem. The comparison of the retrieval time with the temporal resolution of AERI spectrum has been discussed.

**For your convenience, the corresponding revised part in our revised submission is given as follows:**

The average retrieval time of Fast AERIoe for the 826 cases used in the study is 3.69 min, which is more than 50% shorter than that of AERIoe, with an average retrieval time of 8.96 min, which is beyond the temporal resolution (about 8 min) of AERI observations. All of the samples of AERIoe consumed more than 8 minutes, while only 10 cases exceeded the temporal resolution of AERI for Fast AERIoe algorithm. Note that the retrieval time is dependent on the computing platform and the method used to compute Jacobians and are not intended to be directly applied by the reader.

Minor comments:

C5.For the title, may be "Ground-based infrared hyperspectral retrievals of temperature and humidity profile based on Adaptive Fast Iterative Algorithm" is better.

R5: Thanks for the suggestion. The title has been modified as follows:

Retrieval of temperature and humidity profiles from ground-based high-resolution infrared observations using an

adaptive fast iterative algorithm

C6.Line10: "due to" is usually not placed at the beginning of a sentence

R6: Thank you for pointing out the problem. The sentence in line10has been rephrased.

**For your convenience, the corresponding revised part in our revised submission is given as follows:**

Various retrieval algorithms have been developed for retrieving temperature and water vapor profiles from the Atmospheric Emitted Radiance Interferometer (AERI) observations. The physical retrieval algorithm, named AERI Optimal Estimation (AERIoe), outperforms other retrieval algorithms in many aspects except the retrieval time, which is significantly increased due to the complex radiative transfer process

C7.Line12: "part" -> "step"; Line15: "is" -> "was"; Line17: suggest revising to "resulting in an average retrieval time reduction from 8.96 min to 3.69 min" instead of "with the average retrieval time reduced from 8.96 min to 3.69 min"; Line41: "FTIR" -> "The FTIR instrument"; Line45: "which is more advantageous" can be revised to "which makes it more advantageous".

R7: Thanks for the constructive suggestions to improve our manuscript. We have modified the manuscript as suggested by the reviewers.

**For your convenience, the corresponding revised part in our revised submission is given as follows:**

The calculation of the Jacobian matrix is the most computationally intensive step of the physical retrieval algorithm.

The performance of the algorithm was evaluated using synthetic ground-based infrared spectra observations.

... resulting in an average retrieval time reduction by 58.82%.

The FTIR instrument observes near-infrared and mid-infrared high-resolution solar spectra, which are mainly used to retrieve water vapor ...

... which makes it more advantageous in detecting thermodynamic profiles.

C8.Line57: this sentence should be reworked

R8: Thanks for the suggestion. The sentence in line57 has been rephrased.

**For your convenience, the corresponding revised part in our revised submission is given as follows:**

However, the AERIprof algorithm has several significant drawbacks, such as its high dependence on the first-guess profile and inability to provide uncertainty estimates for retrieval results.

**Referee #2**

Dear Referee:

We would like to express our sincere appreciation for your professional review work and valuable comments that greatly helped to improve our manuscript. After deep consideration of your valuable comments, we substantially modify the manuscript. Corresponding changes are highlighted within the document by using smaller font size. Point-by-point responses to your comments are seriously completed for your consideration. If there are still severe issues with our manuscript, please let us know, and we will try our best to modify our article. Thanks for your time!

Overall:

C1. The authors state that the new method results in equivalent retrieval accuracy. However, for water vapor, the bias increase appears to be up to about 40% (0.7 to 0.9), and the RMS increase up to

~12%. This does not seem to me to be comparable retrieval accuracy.

R1: Thank you for pointing this problem out.We agree with the referee that retrieval accuracy of the new method is not comparable to that of AERIoe. However, we believe that the retrieved profiles rather than the retrieval accuracy of the two algorithms are comparable. This is because the increment of BIAS is much smaller than the value of the retrieved water vapor profile itself, and this increment had little impact on the retrieval results of AERIoe. We have modified the statement as "The retrieval results of the fast retrieval model are comparable to that of AERIoe.". Details for this modification are discussed in the response to comments on lines 346 - 348.

C2. The authors need to clarify the scope of the work and fix the errors, typos and unclear parts of the paper. Is the novely of this work just in implementing the k_index and updating the Jacobian less often, or did they also introduce different formulations, methodology, etc? Given that this work follows closely from Turner and Lohnert (2014), the authors should make it clear what is the same as in that prior work by referencing it as needed, avoiding repeating details from it except as necessary, avoiding typos/errors when they do paraphrase from that work, and discussing clearly what is novel in this work. For example, Eqn (1) differs from prior work (and from Rodgers (2000)) in that Xa is replaced with X0. Is this intentional, and if so, why? More examples of specific issues follow.

R2:Thanks for the constructive suggestions to improve our manuscript.Sections 3.1 and 3.2, which are similar to the work from Turner and Lohnert (2014) , have been removed from the manuscript and the descriptions of Fast AERIoe configurations that differ from AERIoe have been added in the paragraph between line142 and line153.

The variable representing the prior profile in Equation (1) in the manuscript is incorrect, and we are really sorry for our careless mistakes. We have replaced $\mathbf{X}_0$ with $\mathbf{X}_a$ in Eqn (1).

Other comments:
C3. Did the authors modify the AERIoe code itself or did they develop a new code base from scratch? Please state in the Data availability if/where/how the fast AERIoe code is available. (Proprietary or open source? How does one obtain it?).

R3: Thank you for the comments.The code for the Fast AERIoe algorithm was developed by ourselves written in MATLAB language. The code for recalculating Jacobians are not publicly available at this time but may be obtained from the authors upon reasonable request.

**For your convenience, the corresponding part in our revised submission is given as follows:**

Data availability. The data used in the manuscript (including AERI, radiosonde, etc) are available from the ARM Data Archive (https://adc.arm.gov/discovery/#/, accessed on 19 January 2022). The code for recalculating Jacobians are not publicly available at this time but may be obtained from the authors upon reasonable request.

C4. Given that the main goal is to reduce the computation time, specifics in that regard are needed. Has the code timing been analyzed and what are the bottlenecks? I assume calculation of the Jacobian is the main bottleneck; is that the case?

R4: We appreciate the suggestion. The code timing has been analyzed both for Fast AERIoe and AERIoe if we understand correctly. The code for the retrieval algorithm was divided into several sections, the time consumed by each section has been analyzed and given in the table bellow,of which the sections denoted with superscript "*" indicate that $\mathbf{K}$ is not recalculated during Fast AERIoe retrieval process.

| Sections | AERIoe | Fast AERIoe |
|---|---|---|
| preparation | 0.29 | 0.22 |

| | | | |
|---|---|---|---|
| iteration 1 | inversion | 0.29 | 0.22 |
| iteration 2 | recalculation of $F(\mathbf{X})$ | 17.11 | 16.69 |
| | recalculation of $\mathbf{K}$ | 68.76 | 70.27 |
| | inversion | 0.31 | 0.27 |
| iteration 3 | recalculation of $F(\mathbf{X})$ | 17.18 | 17.04 |
| | recalculation of $\mathbf{K}$ | 70.55 | 0.00 |
| | inversion | 0.22 | 0.22 |
| iteration 4 | recalculation of $F(\mathbf{X})$ | 17.71 | 16.36 |
| | recalculation of $\mathbf{K}^*$ | 70.07 | 0.00 |
| | inversion | 0.25 | 0.21 |
| iteration 5 | recalculation of $F(\mathbf{X})$ | 16.97 | 17.38 |
| | recalculation of $\mathbf{K}^*$ | 68.93 | 0.00 |
| | inversion | 0.21 | 0.25 |
| iteration 6 | recalculation of $F(\mathbf{X})$ | 16.08 | 15.08 |
| | recalculation of $\mathbf{K}^*$ | 68.23 | 0.00 |
| | inversion | 0.24 | 0.24 |
| iteration final | recalculation of $F(\mathbf{X})$ | 15.91 | 18.45 |
| | recalculation of $\mathbf{K}^*$ | 68.11 | 0.00 |
| | inversion | 0.28 | 0.23 |

From the table, we can find that the recalculation of $F(\mathbf{X})$ and $\mathbf{K}$ consumed an immense amount of time in the retrieval process of AERIoe, and the latter is the most time consuming section. Therefore, by reducing the recalculation of $\mathbf{K}$, the retrieval time of Fast AERIoe is greatly reduced compared to AERIoe.

**For your convenience, the corresponding part in our revised submission is given as follows:**

The codes for the retrieval algorithm are written in MATLAB language and runs on a Lenovo Aircross 510P computer, of which the CPU is Intel Core i7-7700 and the operating system is Ubuntu 14.04. To analyze the code timing of the retrieval algorithm, the code was divided into the following sections: preparation, iteration 1, iteration 2, iteration 3,... and iteration final. The preparation section mainly consists of atmosphere construction, observation vector construction and pre-calculated variables importation. The iteration sections include the recalculation of $\mathbf{K}$ and $F(\mathbf{X})$ and the inversion using equation (1). Note that iteration 1 does not need to calculate $\mathbf{K}$ and $F(\mathbf{X})$ because the prior profile $\mathbf{X}_a$ is fixed (mean value of the atmosphere), and the $\mathbf{K}$ and $F(\mathbf{X})$ associated with it are pre-calculated. The time consumed by each section was analyzed both for AERIoe and Fast AERIoe, results for an arbitrarily selected case are provided in Table 2. The recalculation of $F(\mathbf{X})$ and $\mathbf{K}$ consumed an immense amount of time in the retrieval process of AERIoe, and the latter is the most time consuming section. Therefore, by reducing the recalculation of $\mathbf{K}$, the retrieval time of Fast AERIoe is greatly reduced compared to AERIoe.

**Table 2.** List of time consumption (units: s) by the sections of AERIoe and Fast AERIoe. The sections denoted with superscript "*" indicate that $\mathbf{K}$ is not recalculated during Fast AERIoe retrieval process.

| | Sections | AERIoe | Fast AERIoe |
|---|---|---|---|
| | preparation | 0.29 | 0.22 |
| iteration 1 | inversion | 0.29 | 0.22 |
| iteration 2 | recalculation of $F(\mathbf{X})$ | 17.11 | 16.69 |
| | recalculation of $\mathbf{K}$ | 68.76 | 70.27 |

| | | | |
|---|---|---|---|
| | inversion | 0.31 | 0.27 |
| iteration 3 | recalculation of $F(\mathbf{X})$ | 17.18 | 17.04 |
| | recalculation of $\mathbf{K}$ | 70.55 | 0.00 |
| | inversion | 0.22 | 0.22 |
| iteration 4 | recalculation of $F(\mathbf{X})$ | 17.71 | 16.36 |
| | recalculation of $\mathbf{K}^*$ | 70.07 | 0.00 |
| | inversion | 0.25 | 0.21 |
| iteration 5 | recalculation of $F(\mathbf{X})$ | 16.97 | 17.38 |
| | recalculation of $\mathbf{K}^*$ | 68.93 | 0.00 |
| | inversion | 0.21 | 0.25 |
| iteration 6 | recalculation of $F(\mathbf{X})$ | 16.08 | 15.08 |
| | recalculation of $\mathbf{K}^*$ | 68.23 | 0.00 |
| | inversion | 0.24 | 0.24 |
| iteration final | recalculation of $F(\mathbf{X})$ | 15.91 | 18.45 |
| | recalculation of $\mathbf{K}^*$ | 68.11 | 0.00 |
| | inversion | 0.28 | 0.23 |

Abstract:

C5. Please begin with a sentence that more clearly gives the background - something like: "Two methods for retrieving … are physical and statistical retrieval algorithms …"

R5: Thanks for the constructive suggestions to improve our manuscript. We have modified the problem in our revised submission.

**For your convenience, the corresponding part in our revised submission is given as follows:**

Various retrieval algorithms have been developed for retrieving temperature and water vapor profiles from the Atmospheric Emitted Radiance Interferometer (AERI) observations. The physical retrieval algorithm, named AERI Optimal Estimation (AERIoe), outperforms other retrieval algorithms in many aspects except the retrieval time, which is significantly increased due to the complex radiative transfer process.

C6. Line 12: Begins with "Further analysis showed…" but no analysis has yet been discussed. What changes were made to the Jacobians and why was that expected to speed up performance (but didn't)?

R6: Thank you for pointing out this problem.Analysis of the change of Jacobians in the retrieval process and the dependence of AERIoe algorithm on Jacobians has been added in the abstract.

**For your convenience, the corresponding part in our revised submission is given as follows:**

Analysis of the change of AERI observations' information content with Jacobians revealed that the performance of AERIoe algorithm had little dependence on Jacobians. Thus, the Jacobian matrix could remain unchanged when the variation of atmospheric state is small in the retrieval process. This significantly reduces the amount of computation and thus increases the retrieval speed of AERIoe.

C7. The time estimates are not useful without knowing what type of computing platform was used. Perhaps just give the percent improvement. Also, are 3 significant figures warranted here?

R7: Thank you for the suggestion.We have modified the problem in our revised submission.

**For your convenience, the corresponding part in our revised submission is given as follows:**

The retrieval speed was significantly improved compared with the original AERIoe algorithm under the condition that the parameters of the computing platform remain unchanged, resulting in an average retrieval time reduction by 58.82%.

C8. What is meant by "certain impact"?What is meant by "to some extent"? Why not state the

convergence rate of the traditional algorithm?

R8: Thank you for pointing out the problems. The sentence has been reworked and the convergence rate of the AERIoe algorithm has been added in the abstract.

**For your convenience, the corresponding part in our revised submission is given as follows:**

Results based on synthetic observations revealed that the fast retrieval algorithm reached an acceptable convergence rate of 98%, which is slightly lower than the 99.88% convergence rate of AERIoe for the 826 cases used in this study.

C9. The authors say that "The retrieval accuracy of the fast retrieval model is equivalent to that of the traditional algorithm." However, on lines 346-348 differences indicate that the accuracies are not equivalent.

R9: Thank you for pointing this problem out. We have modified the sentence as "The retrieval results of the fast retrieval model are comparable to that of AERIoe." Reasons for this modification are discussed in detail in the response to comments on lines 346 - 348.

**For your convenience, the corresponding part in our revised submission is given as follows:**

The retrieval results of the fast retrieval model are comparable to that of AERIoe.

C10. How is the convergence criteria adjusted to give reliable retrieval results? It was previously stated that the results were equally accurate. Do you mean they are equally accurate when they both converge?

R10: Thank you for the comments.The sentence 'However, reliable retrieval results can still be obtained by adjusting the convergence criteria.' has been removed.

Lines 115-124:

C11. Line 115: If the authors are using X0 = Xa, they should replace X0 with Xa in Eqn (1) so it is consistent with Turner and Lohnert. If not, they should explain this change.

R11: Thanks for the suggestion. We have replaced $\mathbf{X}_0$ with $\mathbf{X}_a$ in Eqn (1).

**For your convenience, the corresponding part in our revised submission is given as follows:**

$$\mathbf{X}_{n+1} = \mathbf{X}_a + \left(\mathbf{K}_n^T \mathbf{S}_e^{-1} \mathbf{K}_n + \gamma \mathbf{S}_a^{-1}\right)^{-1} \mathbf{K}_n^T \mathbf{S}_e^{-1} \times \left(\mathbf{Y}^m - F(\mathbf{X}_n) + \mathbf{K}_n (\mathbf{X}_n - \mathbf{X}_a)\right), \tag{1}$$

C12. Line 116 says "Y is the observed radiance vector, F(X) is the AERI observed spectrum…" Is it rather that Y is the observed radiance vector (from the observed AERI spectrum) and F(X) is the estimate of Y from the forward model calculation? It would also be helpful to define that the background refers to the a priori atmospheric state, if that is the case.

R12: Thanks for the suggestion. We have replaced '$F(\mathbf{X})$ is the AERI observed spectrum' with '$F(\mathbf{X})$ is the computed radiance for $\mathbf{X}$'. The sentence '$\mathbf{S}_a$ it the background covariance matrix' has been changed to '$\mathbf{S}_a$ is the a priori covariance matrix'.

**For your convenience, the corresponding part in our revised submission is given as follows:**

Here, $\mathbf{X}$ is the profile of the atmospheric state to be retrieved, $\mathbf{X}_a$ is the prior profile of the atmosphere, $\mathbf{S}_a$ is the a priori covariance matrix, $\mathbf{Y}^m$ is the observed radiance vector, $F(\mathbf{X})$ is the computed radiance for X,...

C13. Eqn. 1: I'm curious why this formulation is used instead of the Levenberg-Marquardt formulation (Rodgers 2000, Eqn 5.36). How is the behavior the same or different? Carissimo et al. 2005 state that their method is almost equivalent to Levenberg-Marqardt. In Levenberg-Marquardt, increasing gamma decreases the step size and makes the retrieval weighted more toward steepest descent. How is the formulation here the same or different?

R13: Thank you for the comments, we have put our thinking caps on this question and the discussion regarding it is presented as follows:

The iterative equation for AERIoe is as follows

$$\mathbf{X}_{n+1} = \mathbf{X}_a + \left(\mathbf{K}_n^T \mathbf{S}_e^{-1} \mathbf{K}_n + \gamma \mathbf{S}_a^{-1}\right)^{-1} \mathbf{K}_n^T \mathbf{S}_e^{-1} \times \left(\mathbf{Y}^m - F(\mathbf{X}_n) + \mathbf{K}_n(\mathbf{X}_n - \mathbf{X}_a)\right) \quad (1)$$

The equation for the Levenberg-Marquardt method is given as follows (Rodgers, 2000)

$$\mathbf{X}_{n+1} = \mathbf{X}_n + \left(\mathbf{K}_n^T \mathbf{S}_e^{-1} \mathbf{K}_n + (1+\gamma)\mathbf{S}_a^{-1}\right)^{-1} \times \left(\mathbf{K}_n^T \mathbf{S}_e^{-1}\left(\mathbf{Y}^m - F(\mathbf{X}_n)\right) - \mathbf{S}_a^{-1}(\mathbf{X}_n - \mathbf{X}_a)\right) \quad (2)$$

We believe that the comparison of the two methods can be analyzed from two aspects: the retrieval process and the retrieval results.

(1) In terms of retrieval process, Carissimo et al. (2005) state that gama in Eqn (1) dumps the width of the step between two consecutive iterates and leads its direction toward the steepest descent of the cost function. Gama in Eqn (2) is chosen at each step to reduce the cost function and also tends to the steepest decent of cost function. Therefore, the role of gama in Levenberg-Marquardt method is equivalent to that of AERIoe.

However, the values of gama in the two formulas are quite different, as the profiles in Eqn (1) are retrieved by adjusting the a prior profile $\mathbf{X}_a$, while the profiles in Eqn (2) are iterated by adjusting the iterative profile $\mathbf{X}_n$. For example, in the work of Foth and Pospichal (2017) the initial value of gama is 2, and increases by a factor of 10 if the cost function $\mathbf{J}$ has increased and reduces by a factor of 2 if $\mathbf{J}$ has decreased. Therefore, the value of gama in Levenberg-Marquardt method shows a significant difference from AERIoe ($\gamma = 1000, 300, 100, 30, 10, 3, 1,...$).

(2) In terms of retrieval results, gama in Equation (1) needs to be set to 1 in the final step to eliminate regularization errors in the solutions, which gives

$$\mathbf{X}_{n+1} = \mathbf{X}_a + \left(\mathbf{K}_n^T \mathbf{S}_e^{-1} \mathbf{K}_n + \mathbf{S}_a^{-1}\right)^{-1} \mathbf{K}_n^T \mathbf{S}_e^{-1} \times \left(\mathbf{Y}^m - F(\mathbf{X}_n) + \mathbf{K}_n(\mathbf{X}_n - \mathbf{X}_a)\right) \quad (3)$$

It can be seen that $\left(\mathbf{K}_n^T \mathbf{S}_e^{-1} \mathbf{K}_n + \mathbf{S}_a^{-1}\right)^{-1} \cdot \mathbf{K}_n^T \mathbf{S}_e^{-1} \mathbf{K}_n$ is equivalent to $I - \left(\mathbf{K}_n^T \mathbf{S}_e^{-1} \mathbf{K}_n + \mathbf{S}_a^{-1}\right)^{-1} \cdot \mathbf{S}_a^{-1}$, thus Eqn (3) can be written as

$$\mathbf{X}_{n+1} = \mathbf{X}_a + \left(\mathbf{K}_n^T \mathbf{S}_e^{-1} \mathbf{K}_n + \mathbf{S}_a^{-1}\right)^{-1} \mathbf{K}_n^T \mathbf{S}_e^{-1}\left(\mathbf{Y}^m - F(\mathbf{X}_n)\right) + \left(I - \left(\mathbf{K}_n^T \mathbf{S}_e^{-1} \mathbf{K}_n + \mathbf{S}_a^{-1}\right)^{-1} \mathbf{S}_a^{-1}\right)(\mathbf{X}_n - \mathbf{X}_a) \quad (4)$$

So the solution becomes

$$\mathbf{X}_{n+1} = \mathbf{X}_n + \left(\mathbf{K}_n^T \mathbf{S}_e^{-1} \mathbf{K}_n + \mathbf{S}_a^{-1}\right)^{-1} \times \left(\mathbf{K}_n^T \mathbf{S}_e^{-1}\left(\mathbf{Y}^m - F(\mathbf{X}_n)\right) - \mathbf{S}_a^{-1}(\mathbf{X}_n - \mathbf{X}_a)\right) \quad (5)$$

Eqn (5) is a special case of Eqn (2) when gama in Eqn (2) is chosen to be 0. Therefore, the Levenberg-Marquardt is a combination of a Gauss-Newton (without gama) and steepest descent minimization technique and equivalent to AERIoe.


C14. Figure 1: This figure needs improvement and explanation. E.g. please define "iterative

observations" and "iterative profiles" in the caption. Use of the symbol "Sa" is inconsistent with use of "Jacobians" instead of "K". K_Index has not yet been defined.

R14: Thanks for the suggestion.We have changed "$\mathbf{S}_a$" to "a prior covariance matrix". A symbol "compute monitoring index" was used instead of "K_Index". We have defined "iterative profiles" as temperature and water vapor profiles at iteration $n$ and "iterative observations" as computed radiance for $\mathbf{X}_n$ in the caption.

**For your convenience, the corresponding part in our revised submission is given as follows:**

[Figure]

**Figure 1.**Flowchart of the Fast AERIoe retrieval process. Note that the red line indicates the Jacobian updating process. The iterative profiles and observations are defined as temperature and water vapor profiles at iteration $n$ and computed radiance for $\mathbf{X}_n$. The monitoring index is used to derive the variations of $\mathbf{X}_n$.

C15. Line 118: I don't think n is the number of iterations, but rather the iteration number.

R15: Thanks for the suggestion.We have changed 'the number of iterations' to 'the iteration number'.

**For your convenience, the corresponding part in our revised submission is given as follows:**

... $\mathbf{S}_e$ is the observation error covariance matrix, and $n$ denotes the iteration number ...

C16. Line 120: The description of how gamma is used is not clear.

R16: Thanks for the suggestion.The description of how gamma is used has been added in the revised manuscript.

**For your convenience, the corresponding part in our revised submission is given as follows:**

AERIoe begins with scalar $\gamma$ of large values is to stabilize the retrieval and ends with a unity $\gamma$ to add more information from AERI observations as $n$ increases. This approach allows the AERIoe algorithm to overcome a poor first guess and achieve a results that have the most information from AERI observation, the detailed description of how $\gamma$ is used can be found in Turner and Löhnert (2014).

C17. Line 122: Remove "progress".

R17: Thanks for the suggestion.The word 'progress' in Line 122 has been removed.

**For your convenience, the corresponding part in our revised submission is given as follows:**

As $\gamma$ decreases with iterations, more observation information is introduced to improve the retrieval accuracy.

C18. Line 122: Please change "is not allowed to converge until…" to "Iterations are continued until…" if that is what is meant here.

R18: Thanks for the suggestion. We have modified this problem in our revised manuscript.

**For your convenience, the corresponding part in our revised submission is given as follows:**

Iterations are continued until $\gamma$ decreases to 1 and the following convergence criterion is satisfied.

C19. Line 124: Use consistent symbols. You have superscript n sometimes and subscript n other times.

R19: Thanks for the suggestion. We have changed all of superscript n into subscript n.

**For your convenience, the corresponding part in our revised submission is given as follows:**

$$convergence\_index = \frac{(\mathbf{X}_n - \mathbf{X}_n)\mathbf{S}^{-1}(\mathbf{X}_n - \mathbf{X}_{n+1})}{N} \leq 1, \tag{2}$$

$$K\_Index = \frac{(\mathbf{X}_n - \mathbf{X}_n)^T (\mathbf{X}_n - \mathbf{X}_n)}{N}, \tag{7}$$

C20. Line 214-215: It is not true that "what affects IC and DFS lies only in gamma and Jacobian". In fact, when gamma = 1, IC and DFS are determined by Se and Sa, with the purpose of the Jacobian being to transform Se into the state space for Sa, so that they have the same units and size (rows and columns). I think what you mean is that IC and DFS only change with iteration due to changes in gamma and the Jacobian. (But see below).

R20: We appreciate the suggestion. The sentence in line 214-215 has been rephrased.

**For your convenience, the corresponding part in our revised submission is given as follows:**

It can be seen from equations (3) and (4) that *SIC* and *DFS* are determined by $\mathbf{S}_e$, $\mathbf{S}_a$, $\mathbf{K}$ and $\gamma$. However, $\mathbf{S}_a$ and $\mathbf{S}_e$ remain unchanged during retrieval, which makes *SIC* and *DFS* change with iteration due to variations in $\gamma$ and K. As $\gamma$ drops to 1 at the final iteration, the values of SIC and *DFS* are only dependent on $\mathbf{K}$.

C21. Lines 214 - 232: I don't understand the logic here. On line 224, it is strange to say that gamma changes with the adjustment of the profile, since gamma is prescribed. Figure 3 is confusing. The x-axis goes in the reverse direction as the retrieval proceeds, the figure caption description seems to be wrong (red is actually K_index), and it isn't stated where K_index starts and ends (starts at the high end, ends at the low end?). It is not surprising that the DFS and IC increase as gamma drops to 1, since gamma weights the retrieval away from the observation and toward the first-guess, which presumably has no information content at all. It is also not surprising that there is not much change in DFS and IC with the Jacobian, since, as stated previously, the purpose of the Jacobian here is to transform Se onto the dimensions of Sa. I don't see how this shows that the change of the Jacobian has less influence on the retrieval ability than gamma. Gamma is not supposed to influence the retrieval ability, but only the retrieval stability. That is why iterations are continued until gamma is 1, whereupon the retrieval equation is equivalent to the Gauss-Newton formulation and the maximum information content is used. In fact, I don't see the point of this paragraph or figure at all. The authors could simply state that if X is not changing much, as evidenced by the K_Index, then the Jacobian is probably not changing much either, and therefore does not need to be recomputed. (Note, however, that this is not necessarily true, and they need to show that it is an ok approximation).

R21: We appreciate the suggestion. We admit that the logic here is indeed a bit confusing. Some modifications have been made to Fig 3 and the analysis of this picture. The indicator "*IC*" was changed to Shannon Information Content (*SIC*) to be consistent with the work described by Turner and Löhnert (2014). The change of K_Index with factor γ was removed since it may lead to some confusion and occasional misunderstanding. Fig 3 and the analysis of this figure were reworked.

Thanks for the suggestion again. We added the detailed discussion in the revised manuscript.

**For your convenience, the corresponding revised part in our revised submission is given as follows:**

[Figure]

Figure 2. (a) The change of *SIC* with *K_Index*. (b) The change of *DFS* with *K_Index* for temperature (unfilled circles) and water vapor (open squares), respectively.

The values of *K_Index*, which covers most of the *K_Index* during AERIoe retrieval process (ranged from 0 to 260, see Fig. 3), were obtained by multiplying the a prior profile by different scale factors. The atmosphere dependent **K** were computed by LBLRTM with the prior profiles above, and *SIC* and *DFS* were calculated using equations (3) and (4) with different Jacobians, respectively. Fig. 2 shows the curve of *SIC* and *DFS* changed with *K_Index*. Both of them change slowly with *K_Index*, with the variation of *SIC* within 13.46% (from 13.89 to 16.05), and *DFS* within 4.38% (from 3.71 to to 3.88) for temperature and within 12.73% (from 1.44 to 1.65) for water vapor, which demonstrates that *SIC* and *DFS* remain almost unchanged as *K_Index* increases on the condition that the value of *K_Index* is small. This provides an effective means to improve the retrieval speed of AERIoe by recalculating **K** selectively when **X** is not changing much or *K_Index* is small. This could be achieved by comparing the value of *K_Index* and its threshold at each iteration to determine whether **K** is recalculated or not.

C22. Turner and Lohnert state that "Future versions of AERIoe will use the Carissimo et al. (2005) approach in order to more efficiently converge and reduce computational time." Did the authors explore that approach, and how might that change their analysis?

R22: Thank you for the comments. In fact, the first approach we studied to reduce the retrieval time happens to be the L-curve method recommended by Turner and Lohnert (2014). In our study the codes in Regularization Tools developed by Per Christian Hansen were used to calculate the L-curve and locate the corner, and the retrieval results using gama from the L-curve method did not show any superiority over the method used in AERIoe. The reason lies in that the gama obtained from L-curve method does not gradually decrease with iterations in the retrieval process. In order to figure out the impact of gama on the iterations of AERIoe, we find that the change of **K** is negligible during most of the retrieval process, which inspired this study to reduce the retrieval time by recalculating **K** selectively.

C23. Sections 3.1 and 3.2. are unclear. The description of the retrieval forms is confusing. Is the state vector comprised of the temperature and log of water vapor on the 37 atmospheric layers? Why isn't it parametrized, given that there are far fewer degrees of freedom? It continues to be difficult to tell what is new here and what is the same as previous work. Please avoid repeating details where you

could reference the previous work. For example, you could say, "The forward model is the same as that described by Turner and Lohnert, except as follows…" Was LBLRTM used here to apply the spectral response function, in contrast to the previous work? It is stated that LBLRTM can be used to calculate the Jacobian.   Was it used for this purpose? Again, is this a departure from previous work?

R23: Thanks for the suggestion. We have removed Sections 3.1 and 3.2 from the manuscript and added descriptions of Fast AERIoe configurations that differ from AERIoe in the paragraph between line143 and line154.

**For your convenience, the corresponding part in our revised submission is given as follows:**

Note that $\mathbf{K}$ depends on $\mathbf{X}$ used for estimating the Jacobian, which means that $\mathbf{K}$ must be recomputed for every iteration step. The updating of the Jacobians in the above retrieval process requires the calculation of the optical thickness or radiance (intensity)with respect to different atmospheric constituents at each height, which might be computationally expensive depending on the lengths of $\mathbf{X}$ and $\mathbf{Y}^m$ (Maahn et al., 2020). Owing to the constraints of $\gamma$, the decrease of the difference between simulated and observed radiation is not very much in the adjustment of individual iterations to the retrieval profile. At this time, the change in the Jacobian calculated as per the iteration profile is negligible. Backed by the above analysis, a fast iterative algorithm called Fast AERIoe is proposed on the basis of the AERIoe algorithm. The flowchart of Fast AERIoe is shown in Fig. 1, most of the configurations are consistent with AERIoe described by Turner and Löhnert (2014), except some modifications highlighted as follows:

a. atmospheric configurations: The height grid of $\mathbf{X}$ is consistent with AERIoe, but the maximum retrieval height is limited to 3 km. This is done because the variations of $\mathbf{K}$ above 3 km is negligible due to the fact that most of the information in AERI spectrum lies in the lowest 2 km of the atmosphere for temperature and water vapor profiles (Turner and Löhnert, 2014). The cloud properties were excluded from the state vector $\mathbf{X}$, which is beyond the scope of this study. The corresponding priori profile $\mathbf{X}_a$ and the priori covariance matrices represented by $\mathbf{S}_a$ are modified to be consistent with $\mathbf{X}$.

b. observational vector $\mathbf{Y}$: Spectral regions that sensitive to cloud properties were removed from the observational vector $\mathbf{Y}$ to be consistent with the state vector $\mathbf{X}$. Furthermore, additional observations including surface temperature and water vapor were incorporated into the observation vector, details are described by Turner and Blumberg (2019) .

c. Jacobian matrix $\mathbf{K}$: $\mathbf{K}$ is derived from LBLRTM, which is the same as AERIoe except the version (12.8 instead of 12.1). Another modification is that $\mathbf{K}$ is not recomputed to improve the retrieval speed of the algorithm when the variations of the iterative profile $\mathbf{X}_n$ is small.

C24. Line 176: Please remove the statement that LBLRTM is the most accurate forward model or provide a reference for it.

R24: Thanks for the suggestion. The statement that LBLRTM is the most accurate forward model has been removed.

C25. Line 196-197: Please rephrase this: "…determined whether updating or not by monitoring the indicators that can reflect the changes of Jacobian in the iterative process".

R25: Thanks for the suggestion. The sentence in line 196-197 has been rephrased.

**For your convenience, the corresponding part in our revised submission is given as follows:**

Adaptive updating of $\mathbf{K}$ is the key to reduce the calculation amount of the AERIoe algorithm. The Jacobians are dependent on the atmospheric constituents, which means that $\mathbf{K}$ must be recalculated for every iteration step. The question arises as to under what circumstances $\mathbf{K}$ does not need to be recalculated. Therefore, the dependence of the retrieval capability on Jacobians must be analyzed and indicators that reflect the changes of Jacobians should be figured out to determine whether $\mathbf{K}$ recalculated or not.

C26. Line 344: Please change, "with only slight differences in BIAS metrics between 500 m and 1.5 km" to include a quantitative value, such as "with differences within x% to y%"

R26: Thanks for the suggestion. We have rephrased this sentence and the values of BIAS and RMSE were also changed when radiosonde observations were smoothed with **A**.

**For your convenience, the corresponding part in our revised submission is given as follows:**

The Fast AERIoe retrieved temperature profiles shows a negative deviation of 0.05K between 1.0km and 1.5 km and a maximum increase of RMSE within 0.08 K above 1.0 km when compared with AERIoe.

C27. Lines 346 - 348: It seems like "a maximum increase of 0.29 g/kg in BIAS and a maximum of 0.32 g/kg in RMSE" are significant.    The bias increase appears to be up to about 40% (0.7 to 0.9), and the RMS increase up to ~12%. This does not seem to me to be comparable retrieval accuracy. Please clarify.

R27: Thanks for pointing this problem out. We've made two modifications to Fig.9. One is that the radiosonde observations have been smoothed with the averaging kernel **A**, as suggested by first reviewer, to reduce the vertical representativeness errors.

$$\mathbf{X}_{sonde}^{smoothed} = \mathbf{A}\left(\mathbf{X}_{sonde} - \mathbf{X}_a\right) + \mathbf{X}_a$$

Another modification is the transformation of water vapor into the form of log(ppmv). This is due to the fact that the unit of **K** output by LBLRTM is log (ppmv) during the retrieval, which means that the adjustment of the iterative profile is also in the form of log (ppmv). Therefore, we believe that it is more reasonable to compare water vapor profiles of different retrieval algorithms in the form of log(ppmv).

In the figure of BIAS and RMSE, the Fast AERIoe retrieved temperature profiles shows a negative deviation of 0.05 K between 1.0 km and 1.5 km and a maximum increase of RMSE up to 13.3% (from 0.60 K to 0.68 K) above 1.0 km when compared with AERIoe. For the water vapor profile, the BIAS and RMSE profiles of Fast AERIoe are in good agreement with AERIoe, except for a maximum increase of BIAS up to 25% (from 0.12 log(ppmv) to 0.15 (ppmv)) bellow 1.0 km. Therefore, we agree with the reviewers' comments that Fast AERIoe does not reach a comparable retrieval accuracy. However, we believe that the retrieved profiles rather than the retrieval accuracy of the two algorithms are comparable. This is because the increment of BIAS (within 0.03 log(ppmv)) is much smaller than the value (roughly on the order of 5-10 log(ppmv)) of the retrieved water vapor profile itself, and this increment had little impact on the retrieval results of AERIoe. The comparison of the profiles retrieved by the two algorithms can be demonstrated more clearly by the modified Taylor plots. Most of the blue and red symbols '×' in the figure , which indicate the scores for the individual profiles of the two algorithms, are closed to each other. Therefore, the retrieval results of the AERIoe and Fast AERIoe algorithms are comparable both for temperature and water vapor profiles.

[Figure]

To be more precisely, we have rephrased all the sentence like "the retrieval accuracy of Fast AERIoe is comparable to that of AERIoe" in the manuscript to "the retrieved profiles of Fast AERIoe is comparable to that of AERIoe.

**For your convenience, the corresponding part in our revised submission is given as follows:**

Traditional methods used to evaluate the accuracy of retrieved profiles against radiosondes compute the BIAS and Root Mean Square Error (RMSE), with the calculation formula as follows:

[revised manuscript text omitted]

C28. Line 350: More detail is needed about how you calculated "Pearson's correlation coefficient between two datasets on the y-axis and the ratio of the standard deviation on the x-axis", and the caption of Fig. 10 needs to be improved.

R28: Thanks for the suggestion. The details to calculate the variables in the modified Taylor plot have been added in the revised manuscript. The caption of Fig. 10 has been modified and the sentence 'The intersection of the arms represents the location of the median correlation coefficient and standard deviation ratio of the datasets.' in the caption of Fig.10 has been removed.

**For your convenience, the corresponding part in our revised submission is given as follows:**

Each retrieval/sonde pair is used to derive the correlation coefficient (r) from Eq. (11) and the ratio of the standard deviations from Eq.(12), both are used by Turner and Löhnert (2014).

$$r = \frac{\frac{1}{N}\sum_{z=0}^{z=h}[s(z)-\bar{s}][a(z)-\bar{a}]}{\sigma_s \sigma_a}, \tag{11}$$

$$SDR = \sigma_a / \sigma_s, \tag{12}$$

Within the equations, $s(z)$ and $a(z)$ are defined as the radiosonde observations and retrieved profiles between 0 and 3 km, $(\bar{s}, \bar{a})$ and $(\sigma_s, \sigma_a)$ are the mean values and standard deviations at the same height range.

[Figure]

**Figure 9.** Modified Taylor plots showing the correlation coefficient and standard deviation ratio between the smoothed radiosondes and the retrieved clear-sky (a) temperature and (b) water vapor using AERIoe (red symbols) and Fast AERIoe (blue symbols).There are 826 cases from the SGP site within 2012. Each symbol indicates the score for an individual profile. The arms of the plotted crosses span the 10th–90th percentiles for the correlation coefficient (vertical arms) and the standard deviation ratio (horizontal arms).

C30. English grammar and clarity: If possible, please have a native English speaker edit your paper throughout, including use of "the" in English, which is very challenging to get right. Examples:

Line 25 and throughout: When you are talking about something in general, omit the word "the" and change the noun to the plural. Examples: change "the observation network" to "observation networks". Change "the convective scale numerical weather prediction system" to "convective scale numerical weather prediction systems". Change "the radiosonde profiles" to "radiosonde

profiles". Only use "the" if you are talking about a specific thing, and if you have made it clear which one you are talking about. For example, on line 73, add "the" before "ARM program" since it is clear which program you are talking about (ARM).

R30: Thank you for the suggestion. We have modified the issues identified by the authors in the revised manuscript. Similar issues have been modified and marked in red in the revised manuscript. Moreover, we have carefully polished our paper to improve the English writing. If there are still writing problems with our manuscript, please let us know, and we will try our best to revise our article.

**For your convenience, some modifications in our revised submission are given as follows:**

The accuracy of the initial field provided by observation networks is becoming a key factor restricting the skill of ...

The existing observation networks are insufficient to meet the needs of convective scale numerical weather prediction systems, ...

As the spatiotemporal resolution is too coarse, radiosonde profiles cannot capture the atmospheric phenomena in detail.

The data used in the study are from the ARM program supported by the U. S. Department of Energy, ...

C31. Line 26: Please define all acronyms (e.g. NWP model)

R31: Thank you for the suggestion. We have checked our manuscript carefully and modified the similar problems in our revised submission.

**For your convenience, the corresponding part in our revised submission is given as follows:**

The accuracy of the initial field provided by observation networks is becoming a key factor restricting the skill of numerical weather prediction (NWP) models

C32. Line 32: Change "shows" to "has" or "demonstrates".Line 57: remove "to boot".

R32: Thank you for the suggestion. We have modified the problems in our revised submission.

**For your convenience, the corresponding part in our revised submission is given as follows:**

Space-based detection equipment observes atmospheric upwelling radiance, which demonstrates some drawbacks in the detection of the planetary boundary layer (PBL) owing to the influence of the cloud layer and underlying surface.

However, the AERIprof algorithm has several significant drawbacks, such as its high dependence on the first-guess profile and inability to provide uncertainty estimates for retrieval results.

We are very grateful for your and referees' professional work earnestly. In all, we found the referees' comments are quite helpful. They point the technical issues about our manuscript, also the aspects that we have not done enough. We have tried our best to improve the manuscript and made extensive modifications in the original manuscript according to the comments. Here did not list all the changes but marked in red in revised manuscript.

Thank you and the referees again for your help.

---

## Author Response (AR2)

**Response to EGUSPHERE-2023-637 Review**

Wei Huang , Lei Liu , BinYang , Shuai Hu , Wanying Yang , Zhenfeng Li , Wantong Li and XiaofanYang

Dear editor,

Thank you very much for giving us an opportunity to revise our manuscript. We appreciate the editor and reviewer very much for their constructive comments and suggestions on our manuscript entitled "Retrieval of temperature and humidity profiles from ground-based high-resolution infrared observations using an adaptive fast iterative algorithm" (ID: EGUSPHERE-2023-637). We sincerely thank the editor and reviewer for their valuable feedback that we have used to improve the quality of our manuscript.

We have studied comments carefully and have made extensive modifications which are marked in red in our revised submission manuscript. We have addressed the issues pointed out by the referee, to which we respond in italicized font below. We hope that the referee will find our responses satisfactory, and we are willing to further revise the manuscript regarding any additional suggestions that the referee may have. Please find below the referee's comments in light blue with our responses after each comment.

Kind regards.
Wei Huang
E-mail: whuang_edu@outlook.com

Corresponding author : Lei Liu
E-mail: liulei17c@nudt.edu.cn

**Referee**

Dear Referee:

We would like to express our sincere appreciation for your careful reading and invaluable comments to improve this paper. We have studied carefully on these detailed comments. They are helpful to improve our manuscript. After deep consideration of your valuable comments, we substantially modify the manuscript. Corresponding changes are marked in red in the modified manuscript. Point-by-point responses to your comments are seriously completed for your consideration. If there are still severe issues with our manuscript, please let us know, and we will try our best to modify our article. Thanks for your time!

Replies to the reviewer's comments:

1. Significant figures (see, e.g. https://www.britannica.com/science/significant-figures) - Why are so many significant figures shown? In the abstract it says, "an average retrieval time reduction by 58.82%". Unless this level of precision is somehow important, perhaps just say 59%. For "Convergence rate of 98.67% … slightly lower than 99.88%," perhaps this could be 98.7% vs 99.9%. Similarly for numbers on lines 192-193: use 13% instead of 13.46% etc.

Thank you for your suggestions. We agree with the reviewer that there are too many significant figures shown in the paper. We have adjusted the significant figures throughout the paper and rounded the numbers to 1 digit past the decimal when the precision of the number is not important.

**For your convenience, the modifications to the issues concerned by the reviewer in our revised submission are given as follows:**

*The retrieval speed was significantly improved ..., resulting in an average retrieval time reduction of 59%.*

*... the fast retrieval algorithm reached an acceptable convergence rate of 98.7%, which is slightly lower than the 99.9% convergence rate of AERIoe for the 826 cases used in this study.*

*Both SIC and DFS change slowly with K_Index, as shown in Fig. 2, with the variation of SIC within 13% (from 13.9 to 16.1) and DFS within 4% (from 3.7 to 3.9) for temperature and within 13% (from 1.4 to 1.7) for water vapor,...*

2. The paper still needs to be edited for grammar and clarity in many places.

Thank you for your suggestions. According to your advice, this manuscript was edited for proper English language, grammar, punctuation, spelling, and overall style by one or more of the highly qualified English speakers. We tried our best to improve the manuscript and made some changes in the manuscript. These changes will not influence the content and framework of the paper. Here, we did not list the changes but marked them in red in the revised paper. We earnestly appreciate the Editor/Reviewer's warm and professional work and hope that the corrections will meet with approval.

3. A few points about how their work intersects with the original AERIoe algorithm are unclear to me:

(1) It would be good to state in the paper if/how the reader can obtain the original AERIoe code. I cannot find this in the referenced papers. Does it come with the AERI instrument? Is it proprietary? Can it be purchased?

Thank you for the comments. Unfortunately, we were also unable to obtain the algorithm on the AERI device. The original AERIoe code used in the manuscript was re-written by ourselves on the basis of the work described by Turner and Löhnert (2014).

(2) What programming language is the original AERIoe code written in? C,C++, etc?

To our knowledge, the AERIoe code on the AERI instrument is written in C++ programming language.

(3) How did the authors create their Fast-AERIoe code? Did they modify the original code? Or re-write it?

Thank you for the comments. The Fast AERIoe code was created by modifying the AERIoe code. In this way, the interference of other factors can be excluded and the influence of Jacobian matrix can be highlighted.

(4) The author's state that their new code is written in Matlab. Matlab is a non-compiled code that is not computationally fast. I assume that this is ok because the bottlenecks are computing the Jacobians, and that is done by LBLRTM, which is compiled Fortran code that is therefore fast. Is this the case?

Thank you for the comments. The reviewer is correct. The Jacobians in the Fast AERIoe code are derived from the level analytic Jacobian files in directory AJ, which are created by LBLRTM. The method used to calculate the analytic Jacobians in LBLRTM is faster than the finite differences method.

(5) If so, then I assume LBLRTM writes the Jacobians to files on the disk, which then must be read in by the original AERIoe code or by the Fast AERIoe code. Is this the case? If so, it is possible that

a significant fraction of the time is spent reading and writing from the disk. This could be avoided by calling the Fortran code and passing variables back and forth directly, obviating the need to read/write to disk. I don't know if that can be done with Matlab, but it can be done with Python using f2py. I understand that doing so is probably beyond the scope of this work, but it would be good to know if this is a possible speed-up for future work.

Thank you for the constructive suggestions. The Jacobians need to be recalculated in the retrieval process of AERIoe, which takes up 80% of the total retrieval time. Therefore, reducing the computation of the Jacobian matrix can significantly improve the retrieval speed. However, the time taken to read the Jacobian matrix from the disk is not negligible. The method proposed by the reviewer gives us good enlightenment, and we will explore it in our future work.

Minor edits:

Line 67: change "should be" to "is".

Thank you for the constructive suggestions to improve our manuscript. We have modified the manuscript as suggested by the reviewers.

**For your convenience, the corresponding revised part in our revised submission is given as follows:**

*Additionally, the Jacobian matrix is recalculated for each iteration due to the dependence on the current state vector, which significantly increases the amount of calculation and results in a high retrieval time.*

Line 80: change "will use" to "use".

Thank you for the constructive suggestions to improve our manuscript. We have modified the manuscript as suggested by the reviewers.

**For your convenience, the corresponding revised part in our revised submission is given as follows:**

*We use data collected at the Southern Great Plain (SGP) site, which is located at 36.61 ° N and 149.88 ° W, near Lamont, Oklahoma, USA.*

Line 95: What is meant by redundant data?

Thank you for the constructive suggestions to improve our manuscript. We have corrected "redundant data" to "random error".

**For your convenience, the corresponding revised part in our revised submission is given as follows:**

*AERI has many observation channels, including not only temperature and humidity profile information but also trace gas information such as ozone, methane, and random error.*

Line 167: Should this equation be SIC, not IC?

We sincerely thank the reviewer for the careful reading. As suggested by the reviewer, we have corrected "IC" to "SIC" in equation (3).

**For your convenience, the corresponding revised part in our revised submission is given as follows:**

$$SIC = \frac{1}{2} \ln \det\left(\hat{\mathbf{S}}^{-1}\mathbf{S}_a\right), \tag{3}$$

Line 184: Typo: you forgot n+1. K_index as defined here will always be zero.

Thanks for pointing this problem out. We apologize for our carelessness. In our resubmitted manuscript, the typo has been revised. Thank you for your correction.

**For your convenience, the corresponding revised part in our revised submission is given as follows:**

$$K\_Index = \frac{(\mathbf{X}_n - \mathbf{X}_{n+1})^T (\mathbf{X}_n - \mathbf{X}_{n+1})}{N},$$ (7)

In all, we found the referee's comments are quite helpful. They point the technical issues about our manuscript, also the aspects that we have not done enough. We have tried our best to improve the manuscript and made extensive modifications in the original manuscript according to the comments. Here did not list all the changes but marked in red in revised manuscript. Once again, thank you very much for your constructive comments and suggestions which would help us both in English and in depth to improve the quality of the paper.

Kind regards.
Wei Huang
E-mail: whuang_edu@outlook.com

Corresponding author : Lei Liu
E-mail: liulei17c@nudt.edu.cn

---

## Author Response (AR3)

**Response to EGUSPHERE-2023-637 Review**

Wei Huang, Lei Liu, BinYang, Shuai Hu, Wanying Yang, Zhenfeng Li, Wantong Li and XiaofanYang

Dear editor,

We appreciate the editor very much for his/her constructive comments and suggestions on our manuscript entitled "Retrieval of temperature and humidity profiles from ground-based high-resolution infrared observations using an adaptive fast iterative algorithm" (ID: EGUSPHERE-2023-637). We have addressed the issues noted by the editor, to which we respond in italicized font below. We hope that the editor will find our responses satisfactory, and we are willing to further revise the manuscript regarding any additional suggestions that the editor may have. Please find below the editor's comments in light blue with our responses after each comment.

C1. Line 32: "denser sounding coverage" sounds ambiguous. Geographical coverage or vertical coverage? Please be specific and use the correct adjective.
R1. Thank you for pointing out this problem. We have revised the "denser sounding coverage" to "*wider geographical coverage and higher horizontal resolution*" in line 32 in the improved paper.

**For your convenience, the corresponding revised part in our revised submission is given as follows:**

*Satellite-borne instruments are able to provide wider geographical coverage and higher horizontal resolution than ground-based balloon radiosonde observations.*

C2. Line 49: "radiative transfer process" --> radiative transfer calculation or radiative transfer simulation
R2. We have revised the "radiative transfer process" to "*radiative transfer simulation*" in line 50 in the improved paper.

**For your convenience, the corresponding revised part in our revised submission is given as follows:**

*Physical retrieval algorithms utilize the radiative transfer simulation and the iterative optimization strategy, which exhibit higher retrieval accuracy compared to the statistical retrieval algorithms.*

C3. Line 62: delete "However"
R3. We have deleted the word "However".

C4. Line 63: "strategy of progressively setting regularization parameters from higher to lower values". Do you mean that the regularization parameters are iteration-dependent and are defined as a monotonically decreasing sequence? If yes, please reformulate this sentence and refer this approach (iteratively regularized gauss--newton method) to Bakushinskii (1992) and Xu et al. (2016).
R4. Thank you for your suggestion. The sentence in line 63 has been rephrased. We

have read the references recommended by the editor carefully and added them to the revised sentence.

**For your convenience, the corresponding revised part in our revised submission is given as follows:**

*To achieve good stability and accuracy, the regularization parameters in the AERIoe algorithm are defined as a monotonic sequence, which contains at least seven values, leading to a minimum of seven iterations for convergence because the regularization parameters are iteration-dependent (Bakushinskii, 1992; Xu et al., 2016).*

C5. Line 64: "minimum of seven iterations"? I would guess that you mean maximum number of iterations is 7.

R5. Thank you for your comments. As described in Sect. 3.1, the regularization parameters selected by Turner and Löhnert (2014) come to be a fixed sequence of $\gamma$ factors— 1000, 300, 100, 30, 10, 3, 1, 1, 1, ..., which is used as a function of iteration number. The retrieval is not allowed to converge until $\gamma$ decreases to 1 and meets the convergence criterion, which makes the AERIoe algorithm require at least 7 iterations.

C6. Line 122: To select regularization parameters more wisely, one can choose the initial regularization parameter from the singular values of the Jacobian matrix. Nevertheless, this is more like a suggestion, which may be considered in your future work.

R6. Thank you for the constructive suggestions. The regularization parameter chosen from the singular values of the Jacobian matrix may help the retrieval converge more efficiently and reduce computational time. The method proposed by the editor gives us good enlightenment, and we will explore it in our future work.

Kind regards.
Wei Huang
E-mail: whuang_edu@outlook.com

Corresponding author: Lei Liu
E-mail: liulei17c@nudt.edu.cn